



# CLAAS-3: the third edition of the CM SAF cloud data record based on SEVIRI observations

Nikos Benas[1], Irina Solodovnik[2], Martin Stengel[2], Imke Hüser[2], Karl-Göran Karlsson[3], Nina Håkansson[3], Erik Johansson[3], Salomon Eliasson[3], Marc Schröder[2], Rainer Hollmann[2] and Jan Fokke Meirink[1]

[1]Royal Netherlands Meteorological Institute (KNMI), De Bilt, The Netherlands

[2]Deutscher Wetterdienst (DWD), Offenbach, Germany

[3]Research department, The Swedish Meteorological and Hydrological Institute (SMHI), Norrköping, Sweden

*Correspondence to*: Nikos Benas (nikos.benas@knmi.nl)

**Abstract.** CLAAS-3, the third edition of the Cloud property dAtAset using SEVIRI, was released in December 2022. It is based on observations from the Spinning Enhanced Visible and InfraRed Imager (SEVIRI), on board geostationary satellites Meteosat-8, 9, 10 and 11, which are operated by the European Organisation for the Exploitation of Meteorological Satellites (EUMETSAT). CLAAS-3 was produced and released by the EUMETSAT Satellite Application Facility on Climate Monitoring (CM SAF), which aims to provide high quality satellite-based data records suitable for climate monitoring applications. Compared to previous CLAAS releases, CLAAS-3 is expanded in terms of both temporal extent and cloud properties included, and it is based on partly updated retrieval algorithms. The available data span the period from 2004 to present, covering Europe, Africa, the Atlantic Ocean and parts of South America, the Middle East and the Indian Ocean. They include cloud fractional coverage, cloud-top height, phase (liquid or ice) and optical and microphysical properties (water path, optical thickness, effective radius and droplet number concentration), from instantaneous data (every 15 minutes) to monthly averages. In this study we present an extensive evaluation of CLAAS-3 cloud properties, based on independent reference data sets. These include satellite-based retrievals from active and passive sensors, ground-based observations and in situ measurements from flight campaigns. Overall results show very good agreement, with small biases attributable to different sensor characteristics, retrieval/sampling approaches, and viewing/illumination conditions. These findings demonstrate the fitness of CLAAS-3 to support the intended applications, which include evaluation of climate models, cloud characterization and process studies focusing especially on the diurnal cycle, and cloud filtering for other applications. The CLAAS-3 data record is publicly available via the CM SAF website, at https://doi.org/10.5676/EUM_SAF_CM/CLAAS/V003 (Meirink et al., 2022).



## 1    Introduction

Clouds play an important role in the Earth's atmosphere. They regulate the radiation budget of the Earth-atmosphere system,
assuming different roles in different parts of the radiation spectrum: they act as bright reflectors of the incoming solar
radiation, exerting a cooling effect in the climate system; and they absorb and re-emit infrared radiation, contributing to the
warming of the Earth's surface. Their net effect on the climate system is a slightly cooling one. However, there is high
confidence that the effect of changes in clouds due to global warming is positive (Forster et al., 2021), highlighting the
importance of a continuous monitoring effort.

Monitoring clouds from satellite sensors is a field with several decades of history (Stephens et al., 2019), and known
strengths and limitations: global coverage but infrequent (~twice per day) observations from polar orbiters; continuous
measurements (with frequencies of a few minutes) but regional coverage from geostationary imagers. There are several
cloud Climate Data Records (CDRs) created from passive satellite sensor observations. Well-known examples include the
CDR based on the Moderate Resolution Imaging Spectroradiometer (MODIS, Platnick et al., 2017); the International
Satellite Cloud Climatology Project (ISCCP, Young et al., 2018), which combines retrievals from geostationary and polar
orbiters; the Pathfinder Atmospheres - Extended (PATMOS-x, Foster et al., 2023), which is based on combined data from
the Advanced Very High Resolution Radiometer (AVHRR) and the High-resolution Infrared Radiometer Sounder (HIRS);
the CM SAF Cloud, Albedo And Surface Radiation dataset from AVHRR data (CLARA-A1/2, Karlsson et al., 2013;
Karlsson et al., 2017); and the Cloud property dataset using SEVIRI (CLAAS, Stengel et al., 2014; Benas et al., 2017). The
CLARA and CLAAS CDRs are produced by the Satellite Application Facility on Climate Monitoring (CM SAF), a
consortium established by the European Organisation for the Exploitation of Meteorological Satellites (EUMETSAT).
Cloud-specific properties provided in CLARA and CLAAS include cloud amount, cloud-top pressure and temperature,
optical depth, water path and effective particle radius (liquid and ice).

The third edition of the CLAAS CDR, CLAAS-3, was recently released (Meirink et al., 2022). The two previous editions,
CLAAS-1 and CLAAS-2, have been used extensively, in studies ranging from local scale phenomena to continental scale
analyses and trends, taking advantage of both the high temporal resolution of CLAAS and the continuous, many-year
coverage of a large region of the Earth. Typical examples of CLAAS usage include cloud masking for various applications,
e.g. rain and fog retrieval (Meyer et al., 2017; Egli et al., 2018), study of cloud diurnal and life cycles (e.g. Seelig et al.,
2021; Seethala et al., 2018) and comparisons with model output (e.g. Mallet et al., 2020; Ilić et al., 2022; Amell et al., 2022).

In this study, we provide an overview of the latest edition, CLAAS-3, and present part of a large evaluation effort, which
was completed recently and included various data sets that were used as reference: retrievals from active and passive satellite
sensors, ground observations and in situ measurements from dedicated flights. CLAAS-3 retains all characteristics that
added value and usefulness to CLAAS-2, and includes new, additional features:

- Updated retrieval algorithms, which improve the products and their applicability in relevant studies.



• A longer time series (starting in 2004 and extending to the present), which renders the data set even more suitable for studies of cloud property changes and trends.

• New variables (e.g. the cloud droplet number concentration - CDNC), which expand the scope of studies that CLAAS-3 can support.

The manuscript is structured as follows: in Section 2 we provide a general overview of CLAAS-3, including sensor
properties and calibration (Section 2.1), cloud data sets and file structure (Section 2.2) and the applied retrieval algorithms, where relevant updates are also described (Section 2.3). The data sets used as reference, along with some evaluation approach details, are presented in Section 3. Evaluation results are discussed in Section 4, in groups of relevant cloud properties, and a summary is given in Section 5.

## 2 The CLAAS-3 data record

### 2.1 The SEVIRI sensor

CLAAS-3 is generated based on measurements from the Spinning Enhanced Visible and Infrared Imager (SEVIRI), a passive imaging radiometer with 12 spectral channels ranging from approximately 0.6 µm to 13.4 µm. One of them, the High Resolution Visible (HRV), is a broad-band channel with a nadir resolution of 1 km, as opposed to the 3 km of the other eleven narrow-band channels. From the latter channels, three lie in the visible to shortwave infrared, and the other eight lie in
the thermal infrared part of the spectrum. Details on the SEVIRI channel wavelengths and bandwidths as well as further instrument details are reported in Schmetz et al. (2002). SEVIRI scans the Earth's full disk in a northward direction within 12 minutes. A full scan cycle is repeated every 15 minutes.

SEVIRI operates on the four Meteosat Second Generation (MSG) satellites Meteosat-8, 9, 10 and 11. These are geostationary satellites operated by EUMETSAT, which have been operational in the period from January 2004 to the
present. Their positions have been changed intentionally on several occasions, especially between the 0.0° latitude/longitude and the 9.5° east longitude, which serves the European rapid scan service. Additionally, Meteosat-8 was located at 3.4° west until March 2008. For CLAAS-3, the satellite located over the 0.0° point (or close to it, in the Meteosat-8 case) is always processed, except for data gap cases (e.g. due to sensor decontamination) which are filled from back-up satellites. Thus, CLAAS-3 covers a region of the Earth which includes Africa, Europe and the Atlantic ocean. Parts of South America, the
Indian ocean and the Middle East are also covered close to the edges of this region. Figure 1 shows the Meteosat satellites used for the generation of CLAAS-3 during the 2004-2020 period; Meteosat-11 is also used after 2020.

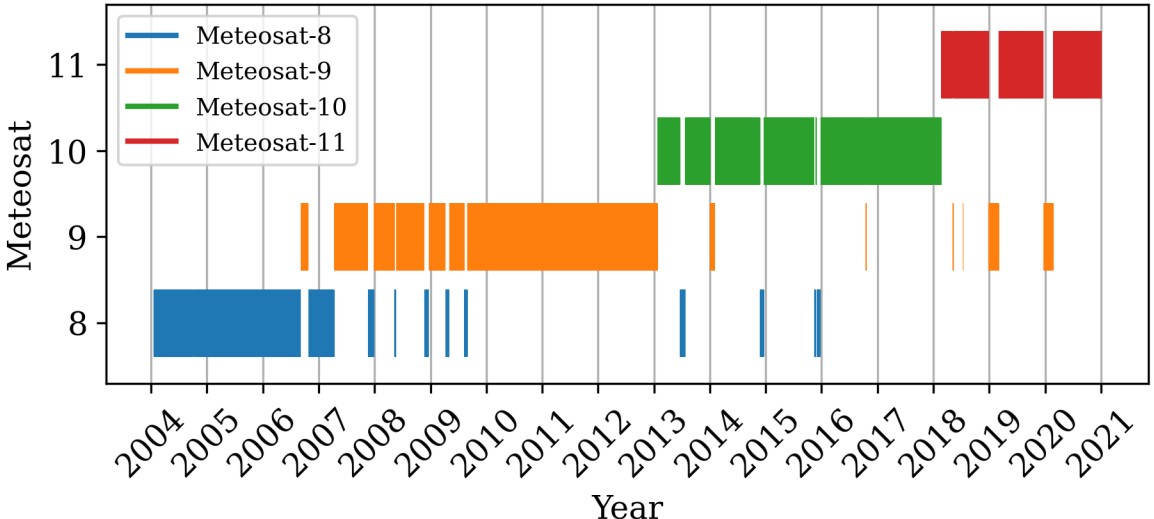

**Figure 1: Overview of the Meteosat Second Generation satellites used for the generation of CLAAS-3. Data gaps are shown enlarged by a factor of 5 for better visibility.**

For CLAAS-3, the three solar channels of SEVIRI (at 0.6 µm, 0.8 µm and 1.6 µm) were calibrated following the methodology described in Meirink et al. (2013), which is based on collocated, ray-matched, atmosphere-corrected, near-nadir reflectances from Aqua MODIS. This approach was updated with the MODIS Collection 6.1 data and extended to include all four MSG satellites. The resulting calibration slopes are available in CM SAF (2022a). For the thermal infrared SEVIRI bands, the EUMETSAT operational calibration slopes were used.

## 2.2 CLAAS-3 contents and structure

### 2.2.1 Cloud properties

CLAAS-3 contains cloud properties that can be grouped in four broad categories, based on the different retrieval algorithms used, which are described in Section 2.3: the CMa/CFC group contains parameters related to initial cloud detection: probabilistic cloud mask (CMa_prob), binary cloud mask (CMa) and cloud fractional coverage (CFC); the CPH group contains cloud phase-related parameters; the CTO group contains cloud-top layer properties, i.e. temperature (CTT), pressure (CTP) and height (CTH); the Cloud Physical Properties (CPP) group contains optical and microphysical properties for liquid and ice clouds, i.e. cloud optical thickness (COT), cloud effective radius (CRE), and secondarily derived cloud water path (separately for liquid and ice water clouds – LWP, IWP), liquid cloud droplet number concentration (CDNC) and cloud geometrical thickness (CGT). Compared to the previous edition, the main differences in terms of content are the inclusion of the new variables CMa_prob, CDNC and CGT.

CLAAS-3 data sets are further categorized into day and night retrievals when possible, i.e. for CMa/CFC, CPH and CTO retrievals. CPP retrievals are available only during daytime, since the retrieval algorithm requires the presence of

observations in the solar channels. Day and night definition in CLAAS-3 is based on solar zenith angle (SZA) thresholds:
SZA < 75° means day, while SZA > 95° is night.

Another data set distinction is based on the Shortwave Infrared (SWIR) channel used in the CPP algorithm retrievals: they are available either based on the 1.6 µm channel or on the 3.9 µm channel. The same holds for CPH, which is refined in the last step of its retrieval, based on the output of CPP. This is an addition compared to CLAAS-2, where only 1.6 µm retrievals were provided. CDNC and CGT are retrieved based on the 3.9 µm channel only. These data set characteristics are further
explained in Section 2.3, where the CLAAS-3 retrieval algorithms are described. Further information is also available in CM SAF (2022b).

Finally, the level 3 monthly mean CFC is additionally available at three cloud layers, (low, middle and high), defined based on their cloud-top pressure. Following the ISCCP conventions, low clouds have CTP higher than 680 hPa, middle cloud CTP lies between 680 and 440 hPa, and high clouds have CTP lower than 440 hPa.

**2.2.2    File structure**

CLAAS-3 data files are organized in processing levels, which is typical for satellite-based retrievals. Following this scheme, level 2 retrievals are provided on an instantaneous basis, every 15 minutes, which is the scanning frequency of SEVIRI. Level 3 provides temporal and spatial averages of the level 2 data. All level 2 variables are provided at the SEVIRI native resolution grid (3 km × 3 km at nadir, expanding towards the disk edge), and level 3 monthly averages are given in a 0.05°
regular grid. Monthly diurnal cycle averages are available at a 0.25° resolution.

In each processing level, CLAAS-3 variables are further grouped in different files based on their broad parameter categorization described before. Thus, in level 2 there are CMA, CPP and CTX files, with CTX including cloud-top layer properties. Files providing data on geometry and illumination conditions (ANG files) are also available upon request, although they are not part of the official data record. In level 3, properties are grouped in files CFC, CPH, CTO, LWP, IWP
and JCH, with the latter including monthly Joint Cloud properties Histograms of CTP and COT at 15 and 13 intervals, respectively.

It should finally be noted that, apart from cloud mask and cloud phase variables, all data come with a grid cell level uncertainty estimation at level 2, which is fully propagated into level 3 aggregations following the approach described in Stengel et al. (2017).

**2.3    Retrieval algorithms used**

Detailed descriptions of the retrieval algorithms used in CLAAS-3 can be found in the respective Algorithm Theoretical Basis Documents, cited below. Here, for each algorithm a brief overview is given.

Cloud detection in CLAAS-3 starts by retrieving a probabilistic cloud mask (CMa_prob) using a Naive Bayesian approximation, where the total cloud probability is estimated by multiplying individual, assumed independent, probabilities.
For the calculation of the latter, the method uses a set of predefined image features to capture variability due to a range of



factors, including solar and satellite geometry, prevailing atmospheric conditions and underlying surface characteristics. The algorithm is trained using collocations of SEVIRI reflectances with cloud observations from the Cloud-Aerosol Lidar with Orthogonal Polarization (CALIOP), which are used as reference. After the calculation of CMa_prob, a threshold of 50% probability is used to distinguish between clear and cloudy scenes, and create the binary cloud mask. All other cloud

properties are retrieved for cloudy pixels based on this distinction, while CFC is defined as the fraction of cloudy pixels per (level 3) grid cell compared to the total number of analyzed pixels in the grid cell. Detailed information on the CMa_prob algorithm is given in Karlsson et al. (2020) and in CM SAF/NWC SAF (2021). Specific algorithm adaptations for the SEVIRI-based CLAAS-3 retrieval are also described in CM SAF (2022c). Compared to CLAAS-2, this algorithm constitutes a major upgrade; cloud detection previously relied on a series of spectral threshold tests which, among other factors,

depended on illumination conditions and surface types (see Benas et al., 2017 and relevant references therein).

For the retrieval of CLAAS-3 CTP, a multilayer perceptron neural network is used. Training of the network is based on a data set of collocations between SEVIRI and CALIOP 5 km data. Network input includes variables derived from the SEVIRI infrared channels at 3.7 μm, 8.5 μm, 11 μm and 12 μm (brightness temperatures and relevant differences) and collocated variables from Numerical Weather Prediction (NWP), including surface pressure, column integrated water vapor and

temperature at specific pressure levels. SEVIRI data coming from the full Meteosat disk cover all cases of variations due to differences in viewing and illuminations conditions. Two neighbouring pixels are included as variables and this significantly improves the results. Uncertainties are estimated a posteriori based on a Quantile Regression Neural Network (Pfreundschuh et al., 2018) for the 16th and 84th percentile. CTP is subsequently converted to CTT and CTH based on collocated NWP profiles. The algorithm is described in detail in Håkansson et al. (2018), with adaptations for SEVIRI given in CM SAF

(2022c). As in the cloud detection case, this algorithm is also a major upgrade compared to the CLAAS-2 retrieval; the latter relied on simulations of radiances and brightness temperatures in various SEVIRI channels, for both cloud and clear sky conditions, and the retrieval of CTP by comparisons with respective observations (see Benas et al., 2017 and references therein).

The retrieval of CPH in CLAAS-3 is based on a series of spectral threshold tests applied in a specific order and involving

SEVIRI channels 4, 5, 7, 9, 10 and 11 (3.9, 6.3, 8.7, 10.8, 12.0 and 13.4 μm, respectively). For the evaluation of infrared channel brightness temperatures, cloudy sky radiance profiles and clear sky brightness temperatures are calculated using the Radiative Transfer for TOVS (RTTOV) model version 11.3, developed within the NWP SAF (Saunders et al., 2018). The CPH retrieval algorithm is an adaptation of a version first applied to AVHRR data (Pavolonis et al., 2005). The algorithm output initially yields six types of clouds: liquid, supercooled, opaque ice, cirrus, overlap and overshooting which are then

grouped in liquid and ice phase (the first two and last four, respectively). Further details can be found in CM SAF (2022b).

Subsequent to the CPH retrieval, the retrieval of cloud optical and microphysical properties (COT, CRE, CWP, CDNC and CGT) with the CPP algorithm is applied. CPP is run only for daytime cloudy pixels, and CPH is used as input to distinguish between liquid and ice clouds and use the appropriate retrieval scheme. For both liquid and ice clouds a Look-Up Table (LUT) is used for the simultaneous retrieval of COT and CRE, using one visible (0.6 μm) and one SWIR (1.6 or 3.9 μm)



channel. This is a widely used method in satellite retrievals, largely based on the work of Nakajima and King (1990). With
       CPP, this method is applied to SEVIRI and AVHRR reflectances (Roebeling et al., 2006). LUT reflectances are calculated
       using the Doubling-Adding KNMI (DAK) radiative transfer model (Stammes, 2001) for horizontally and vertically
       homogenous clouds in a Rayleigh scattering atmosphere, with settings detailed in CM SAF (2022b). LWP and IWP are then
       calculated based on COT and CRE, assuming vertically homogeneous water content (e.g. Stephens, 1978). For liquid clouds,

and for the 3.9 μm retrievals only, CDNC and CGT are additionally retrieved from COT and CRE, assuming an idealized
       stratiform boundary layer cloud, as described in Bennartz and Rausch (2017). The major updates applied in CPP version 6.0,
       used in CLAAS-3, compared to version 5.2, used in CLAAS-2, are the introduction of retrievals using the 3.9 μm channel
       (including the new products CDNC and CGT), and the updated microphysical assumptions regarding liquid clouds (narrower
       droplet size distribution, based on results from Benas et al., 2019) and ice clouds (roughened aggregates of solid columns in

place of roughened hexagonal crystals – the former described in Yang et al., 2013 and in Baum et al., 2011). Additionally,
       uncertainty estimates were revised and extended, and surface albedo ancillary input data were also updated and improved.
       Further details are provided in CMSAF (2022b).

       It should be noted that after 2020 the CLAAS-3 production is based on slightly different ancillary data, in order to facilitate
       production in near real time (a lag of a few days in practice). This part of the time series is called Intermediate Climate Data

Record (ICDR).

## 3    Reference data sets and evaluation scores

### 3.1    CALIOP

CALIOP is an instrument on board the Cloud-Aerosol Lidar and Infrared Pathfinder Satellite Observation (CALIPSO)
satellite, which was launched in April 2006. CALIOP measures the backscatter intensity at 1064 nm and the orthogonally

polarized components of the backscattered signal at 532 nm, providing detailed profile information of aerosol and cloud
       particles, and relevant physical parameters (Winker et al., 2009). The original measurement resolution is 333 m horizontally
       and 30-60 m vertically, with first measurements becoming available in June 2006.

       CALIOP data are used here for the evaluation of level 2 CMa, CPH and CTO, and level 3 CFC and CTO. For the level 2
       comparisons we used the CALIOP level 2, 5 km cloud layer (CLAY) dataset version 4.20 (CAL_LID_L2_05kmCLay-

Standard-V4-20, NASA/LARC/SD/ASDC, 2018). The 5 km resolution refers to the along-track direction and it is
       constructed from several original 333 m footprints. It was selected as the closest to the CLAAS-3 level 2 resolution.
       Additionally, the combination of several original CALIOP profiles in the 5 km product increases the signal-to-noise ratio,
       rendering the identification of thin cirrus clouds more reliable, and the comparison with CLAAS-3 more fair. Collocation of
       CALIOP level 2 retrievals with CLAAS-3 grid cells was based on a nearest neighbor approach in both space and time,

leading to maximum distances of 5 km and 7.5 minutes, respectively. These collocations cover the entire 2013, ensuring
       seasonal representativity. After the collocation, the CALIOP cloud fraction was binarized using a 50% cloudiness threshold.



This is required to allow for a comparison against the CLAAS-3 binary cloud mask. Retrievals known to be of reduced quality, i.e. SEVIRI viewing zenith angles (VZAs) higher than 75° and CALIOP information flagged as bad quality, were excluded from the analysis.

For the level 3 evaluation we used a special CALIOP level 3 product which was created for the GEWEX cloud assessment study (NASA/LARC/SD/ASDC, 2019). It is based on the level 2 CLAY version 4.20 product for the computation of monthly averaged cloud parameters at 1° horizontal resolution. The data set is available in two flavors: the top layer flavor, which takes into account all cloud layers in each profile, and the passive flavor, which chooses exclusively the top layer clouds that would be safely detected by a passive imager, i.e. the ones remaining after removing the upper layer with COT

values equal to or less than 0.3. Both flavors were used for comparisons with CLAAS-3.

Uncertainties in the CALIOP data used here originate mainly in the spatial averaging process, for two reasons. First, it cannot be applied in the across-track direction; the relevant effect is expected to be minor, since most clouds have aspect ratios that ensure detection by CALIOP. Second, there are inconsistencies even among CALIOP data sets with different horizontal resolutions, due to different averaging methodologies (e.g. Karlsson and Johansson, 2013). An additional

uncertainty stems from the slightly different cloud detection efficiency during day and night, that can introduce a difference of less than 1% between day- and night-time data (Chepfer et al., 2010). A more detailed discussion on uncertainties and error sources from CALIOP is included in CM SAF (2022d).

## 3.2 SYNOP

Synoptic observations of total cloud cover from meteorological stations (SYNOP) were used for the evaluation of CLAAS-3

level 3 CFC. SYNOP data came from the Deutscher Wetterdienst (DWD, German Meteorological Service) archive of reports from meteorological stations. These measurements follow the Guide to Meteorological Instruments and Methods of Observations framework (Jarraud, 2008). The data set covers the entire CLAAS-3 period, while the spatial coverage is limited to land, with higher station density in the northern mid-latitudes.

For the creation of a consistent and high quality reference data set based on SYNOP, the following criteria were applied to

observations performed in the SEVIRI disk: a) only manned airport stations were considered, being more reliable and having continuous time series and frequent measurements; b) monthly mean CFC values from SYNOP were computed from at least 20 daily mean values, and daily means were computed from at least six observations; c) stations located beyond 75° of SEVIRI VZA were omitted; d) after applying the above criteria, only stations covering at least 95 % of the period 2004-2020 were included. Out of ~1800 stations in the SEVIRI disk, 504 passed all the above criteria and were used in the comparisons.

For the latter, CLAAS-3 level 3 CFC values were averaged over an area of 5 × 5 grid cells surrounding the SYNOP station in order to reflect the typical spatial extent of CFC observations by human observers.

Uncertainties and errors in SYNOP observations are mostly related to human limitations: subjectivity due to different interpretations, detection limit of the human eye in discerning clouds of low optical thickness and clouds in a night-sky





background, and overestimation of convective clouds at slanted views (Karlsson, 2003). CM SAF (2022d) suggest an overall
accuracy of SYNOP observations between -10% at night time and +10% at daytime conditions.

### 3.3 MODIS

The Moderate Resolution Imaging Spectroradiometer (MODIS) is an advanced imaging sensor flying on board NASA's
Terra and Aqua satellites, since 1999 and 2002, respectively. Both satellites follow a sun synchronous orbit, with equatorial
crossing times at 10:30 (Terra) and 13:30 (Aqua). With its wide swath, MODIS views the entire Earth's surface every one to
two days, acquiring observations in 36 spectral bands.

MODIS-based products are used here extensively, for the evaluation of both level 2 (CPP products) and level 3 (all cloud
parameters) CLAAS-3 products. The main reasons for this selection are the complete coverage of the CLAAS-3 period, the
proven stability of both MODIS instruments and their advanced features: additional shortwave channels allow for better
discrimination of thin cirrus clouds and more reliable optical properties retrieval over bright surfaces.

Both level 2 and level 3 MODIS cloud products used here come from collection 6.1 (Platnick et al., 2017). Level 2 data are
available in files of 5-minute granules, at a spatial resolution of 1 km. Collocation with CLAAS-3 level 2 data was based on
spatial averaging of MODIS data within the SEVIRI pixels for the nearest images in time. Since MODIS level 2 data were
used for the evaluation of CPP products, all MODIS pixels corresponding to a SEVIRI pixel were required to be cloudy and
to have the same phase as the CLAAS-3 retrieval. The same (cloudiness and phase) requirement also applied to the MODIS
pixels directly surrounding the SEVIRI pixel, in order to ensure exclusion of cloud edges from the comparisons. Due to the
high volume of data involved, only March 2013 was considered in the level 2 comparisons. Both Terra and Aqua MODIS
level 2 data were used every $5^{th}$ day, in order to further reduce the large amount of available data. In the case of level 3 data,
the entire time series was analyzed. MODIS level 3 data contain monthly averaged parameters at $1° \times 1°$ resolution. Average
values of Terra and Aqua MODIS were used in the level 3 comparisons, to imitate as well as possible the diurnally averaged
CLAAS-3 level 3 data.

CDNC and CGT are not included in the level 2 MODIS products. They were computed from the COT and CRE retrievals
(based on the MODIS 3.7 μm channel), using the method described in Bennartz and Rausch (2017).

Uncertainties in MODIS retrievals are expected to be similar or slightly smaller than those from the SEVIRI retrievals. The
latter is expected especially when a wider variety of channels is used, e.g. in the thin cirrus clouds discrimination and the
retrieval of cloud optical properties over bright surfaces. As mentioned previously, in such cases additional SW channels
lead to more reliable results.

### 3.4 AMSR2 and MAC-LWP

Microwave (MW) sensors are used for the evaluation of CLAAS-3 LWP in both levels 2 and 3. In level 2 comparisons are
performed against the LWP retrieved from the Advanced Microwave Scanning Radiometer 2 (AMSR2). AMSR2 is a dual
polarization, conical scanning passive MW sensor with 16 channels in the range 6.9-89 GHz. It flies since 2012 on board the





Japan Aerospace Exploration Agency (JAXA) Global Change Observation Mission for Water (GCOM-W) satellite in a sun-synchronous orbit with an equatorial crossing ascending node at 13:30 local time. Among other climate variables, AMSR2 retrieves LWP over ocean with a spatial resolution of 7 km × 12 km, which is aggregated to a 0.25° × 0.25° regular grid, separately for the two nodes. Here, data version 8.2 from the (daytime) ascending node are used (Wentz et al., 2014).

Information on the relevant retrieval algorithms is given in Hilburn and Wentz (2008).

For the comparison of CLAAS-3 level 2 LWP with AMSR2, data from all days of one month (March 2013) were used. In each AMSR2 grid cell (0.25° × 0.25°) roughly 9 × 9 CLAAS-3 pixels (at nadir) are present. Thus, for every valid AMSR2 LWP value, an average CLAAS-3 LWP was estimated, based on the time slot closest to the AMSR2 measurement time, with the additional requirement that all CLAAS-3 grid cells are either liquid cloud or clear sky.

Level 3 CLAAS-3 LWP is compared with data from the Multisensor Advanced Climatology of Liquid Water Path (MAC-LWP) version 1 (Elsaesser et al., 2017). The data set is compiled from retrievals based on various passive MW sensors, including AMSR2 and AMSR-E, the Global Precipitation Measurement Microwave Imager (GMI), the Special Sensor Microwave Imager (SSM/I) series and the Tropical Rainfall Measurement Mission Microwave Imager (TMI). It is available during 1988-2016 at 1° × 1° resolution as monthly averages and on a monthly averaged diurnal (hourly) basis.

Both these options are used here to evaluate the monthly averaged CLAAS-3 time series and its diurnal variability. This evaluation focuses on a region in the southeastern Atlantic, where stratocumulus clouds prevail (0-10°E, 10-20°S). This is because MW-based retrievals are not sensitive to ice clouds, so that regions with as low as possible ice cloud fraction should be selected for a meaningful comparison with optical imagers. The comparison covers the period 02/2004 – 12/2016, when both data sets are available. MAC-LWP provides all-sky LWP, and corresponding CLAAS-3 values were used in both

monthly mean and monthly mean diurnal cycle comparisons. Since these are not directly available in the latter case, they were computed by multiplying the (in-cloud) LWP with the corresponding cloud fraction (CFC) and its liquid portion (CPH). To ensure consistency, only cases when the study area was fully covered by both data sets were selected. The monthly mean CPH was found to be always higher than 92%, confirming the almost exclusive presence of liquid clouds initially assumed.

Error sources in MW-based LWP retrievals include the cloud-rain partition bias, the cloud temperature bias and the cloud fraction-dependent bias (Greenwald et al., 2018). For MAC LWP, Elsaesser et al. (2017) report an error ranging between 10% and 25%, depending on location.

## 3.5 ORACLES

The ORACLES project (Observations of Aerosols above Clouds and their Interactions) investigated processes and effects related with biomass burning aerosols that originate in southern Africa and are transported over the southeastern Atlantic. Additional focus was given to their interactions with the stratocumulus clouds that form large decks over this region (Redemann et al., 2021). During ORACLES, CDNC was measured in 35 flights during September 2016, August 2017 and



October 2018, using a Flight Probe Dual Range Phase Doppler Interferometer. The measurement technique and associated
uncertainties are described in Chuang et al. (2008).

CDNC measurements from these flights are available per second. For the comparison with CLAAS-3 level 2 CDNC, each measurement was first collocated with a specific CLAAS-3 time slot and grid cell, based on aircraft navigation data. Measurements lower than 20 cm$^{-3}$ were set to no-data. This was decided based on a time series analysis that showed that these low values practically don't correspond to detectable clouds. Each group of ORACLES measurements found in the
same CLAAS-3 grid cell and time slot was averaged and the value was added to the collocation data set, only if no-data case was found in the group.

Since the ORACLES flights were dedicated to study biomass burning aerosols, there is a risk that the presence of such aerosols will affect the quality of the CLAAS-3 CDNC retrieval and consequently the comparison results. For this reason, Absorbing Aerosol Index (AAI) data were additionally used to filter cases with high aerosol loads. These data came from
retrievals using the Global Ozone Monitoring Experiment-2 (GOME-2) instrument on board MetOp-A and MetOp-B satellites, and were available via the ESA Aerosol CCI project (Tilstra et al., 2010; ESA Aerosols CCI project team, 2020).

Following recommendations regarding satellite-based retrievals of CDNC, which are discussed in Grosvenor et al., (2018), the following quality criteria were applied in the CLAAS-3 and ORACLES CDNC collocations data set:

- Cases where SZA>60° were excluded, since high SZAs are associated with large biases in COT and CRE retrievals.
- Only cases where CRE retrieved at 3.9 μm is greater than CRE retrieved at 1.6 μm were considered. This in practice confirms that CRE increases toward the cloud top. This assumption is also applied in the Bennartz and Rausch (2017) CDNC data set based on MODIS.
- Cases with AAI>2, indicating high loads of absorbing aerosols that may affect the CLAAS-3 retrieval, were excluded.

Additional filters in minimum COT (exclusion of thin clouds) and number of measurements in the CLAAS-3 grid cells were also tested, with no significant effect on the comparison results.

## 3.6   DARDAR

The DARDAR (raDAR/liDAR)-CLOUD product is used for the evaluation of level 2 IWP and ice CRE. DARDAR uses an optimal estimation approach to retrieve ice cloud properties by synergistically combining measurements from CALIOP,
CloudSat radar and MODIS (Cazenave et al., 2019; Delanoë and Hogan, 2008). The version 3.00 product, used here, is available at 1.1 km horizontal and 60 m vertical resolution.

For the comparison with CLAAS-3, DARDAR profiles were considered in groups of five, and the requirement was applied that all these profiles contain ice cloud and none of them contain layers with liquid cloud or rain. If the requirement was met, the three central profiles were averaged for the estimation of COT, CRE and IWP. In the case of CRE, the estimation was



vertically weighted towards the cloud top, in order to reflect the sensitivity of passive sensor retrievals to this part of the cloud (Platnick, 2000). Specifically, the weighted CRE, $r_e^{\text{top}}$, was calculated following:

$$r_e^{\text{top}} = \frac{\int_0^{\text{TOA}} r_e(z)\alpha(z)\,e^{-\tau(z)/\tau_w}\,dz}{\int_0^{\text{TOA}} \alpha(z)\,e^{-\tau(z)/\tau_w}\,dz}$$

( 1 )

$$\tau(z) = \int_z^{\text{TOA}} \alpha(z')\,dz'$$

( 2 )

where TOA is top-of-atmosphere, $r_e(z)$ the effective radius profile, $\alpha(z)$ the extinction profile, $\tau(z)$ the integrated cloud optical thickness above height $z$, and $\tau_w$ the optical thickness determining how far into the cloud the weighting is applied.

For the creation of CLAAS-3 and DARDAR collocation pairs, the CLAAS-3 pixel closest to the central DARDAR profile was first located and the SEVIRI time slot nearest to this profile measurement time was determined. It was then required that

all 3 × 3 CLAAS-3 pixel retrievals centered on the DARDAR profile are ice clouds, for the minimization of cloud edge effects. If that was the case, then ice COT, CRE and IWP values from the central CLAAS-3 pixel were paired with corresponding DARDAR values and added to the collocation data set.

### 3.7    Cloud detection scores

The methodology used to evaluate CLAAS-3 data sets varies depending on specific characteristics of the data (e.g.

processing level, binary or continuous values) and the data sets used as reference. For level 2 binary data (CMa and CPH – cloudy/clear and water/ice, respectively), detection scores are calculated, to evaluate their performance relative to respective CALIOP data, and to assess their dependence on varying the CALIOP layer that is considered as cloud top.

Detection scores used include Probability Of Detection (POD), False Alarm Ratio (FAR), hit rate and the Hansen-Kuipers Skill Score (KSS). Definitions of these scores are given in Benas et al., (2017). In short:

• POD is the fraction of correctly retrieved instances of one case (e.g. cloudy) relative to all retrieved cases (cloudy and clear in this example). POD range is 0 to 1, with 1 being the perfect score.

• FAR is the fraction of false retrievals relative to all same-value retrievals. FAR range is 0 to 1, with 0 being the perfect score.

• Hit rate is the fraction of all correct retrievals (where CLAAS-3 and reference agree) relative to all possible

combinations of CLAAS-3 and reference retrievals (range is 0 to 1, with 1 being the perfect score).

• KSS shows how well the retrieval separates the two discrete cases (range is -1 to 1, with 1 being the perfect score and 0 indicating no skill).



# 4    Evaluation results

## 4.1    CMa and CFC

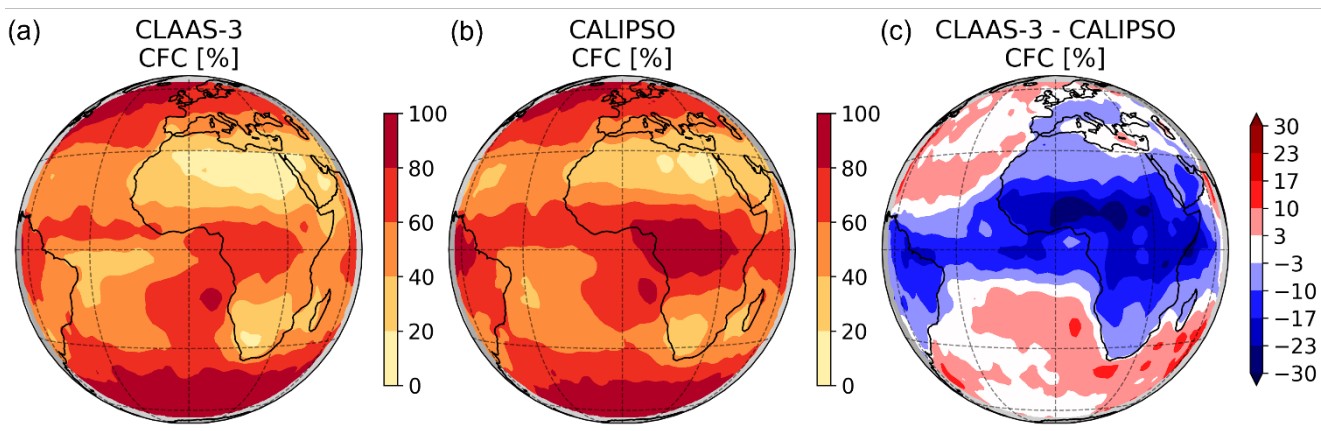


**Figure 2: Spatial distribution of the 2013 average level 2 Cloud Fractional Coverage (CFC) from CLAAS-3 (a), CALIOP (b), and their difference (c). Here CFC is estimated from the binary cloud mask by collecting matchups to a regular 1.5° × 1.5° grid and averaging them within each grid box. CALIOP cloud detection criterion is total column COT > 0. A 2-dimensional Gaussian filtering was used for noise reduction.**

Figure 2 shows the spatial distributions of CFC from CLAAS-3, CALIOP and their difference, estimated by averaging all binary level 2 CMa matchups in 2013 in a regular 1.5° × 1.5° grid. The large-scale patterns are very similar in the two data records. The main differences occur around the Intertropical Convergence Zone (ITCZ), where CALIOP reports a significantly higher cloud fraction, leading to negative biases in the range 10%-30%. This can be explained by the large percentage of thin cirrus clouds in this region, which are more likely to be missed by CLAAS-3. Positive biases at the edge

of the disk should be attributed to CLAAS-3 overestimation of cloud fraction at large viewing angles.

The spatial patterns of the bias shown in Figure 2c show an overall CLAAS-3 underestimation of the cloud presence compared to CALIOP. This feature is expected, since CALIOP is more sensitive to optically thin clouds. As a result, it is also strongly dependent on the CALIOP cloud filtering criterion (considered to be "total column COT>0" in Figure 2). To analyze this sensitivity, comparisons were also performed against CALIOP CMa derived after omitting a non-zero columnar

COT from the cloud top (see, e.g., Karlsson and Håkansson, 2018).

The sensitivity of the CLAAS-3 cloud detection scores to the selected minimum COT threshold is shown in Figure 3. The increase in POD with increasing COT threshold verifies that CLAAS-3 is less sensitive to optically thin clouds. However, the simultaneous increase in FAR indicates that there are also cases of optically thin clouds correctly detected by CLAAS-3. As the COT threshold increases, these cases will gradually switch to clear-sky, leading to an increase in FAR. The

combination of these effects causes the hit rate and KSS to peak at COT = 0.1. This means that CLAAS-3 systematically misses clouds which are optically thinner than approximately 0.1. These results also justify the use of COT > 0.1 as the CALIOP cloud detection criterion in the CPH and CTH comparisons discussed below.

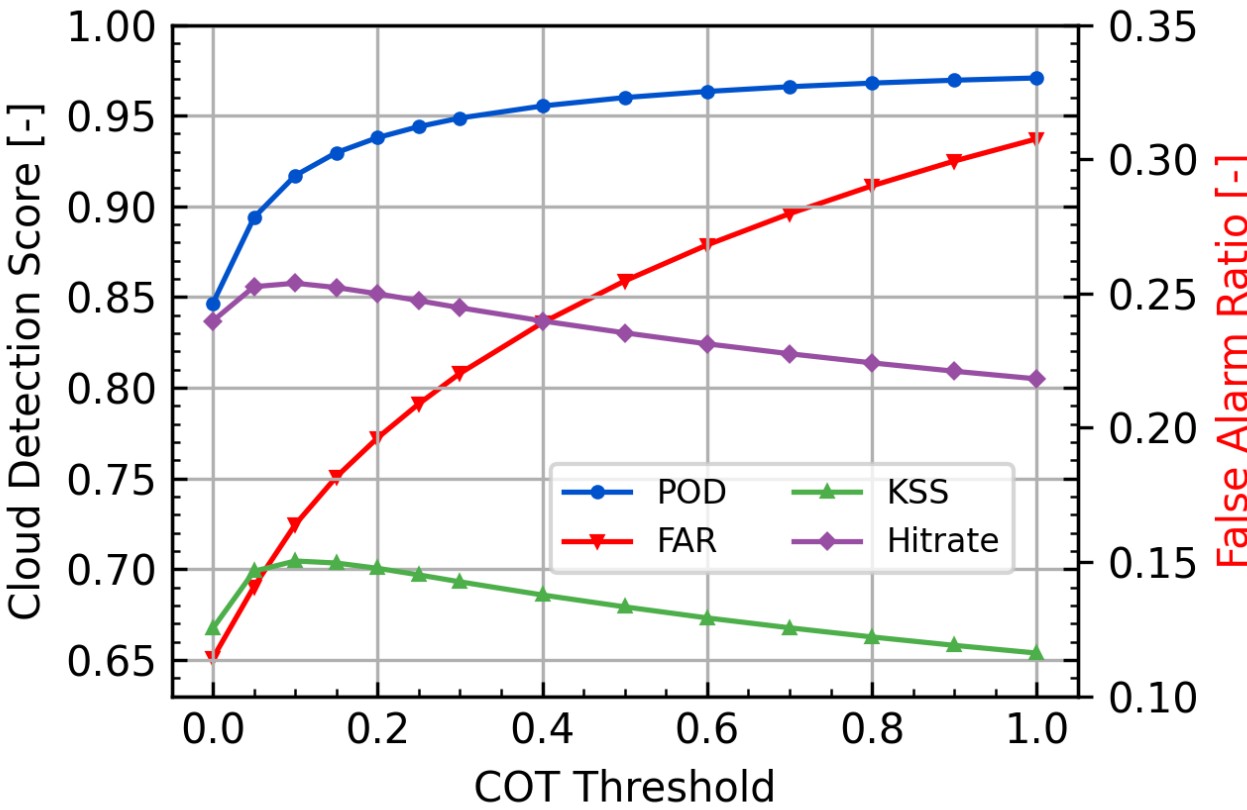

**Figure 3: CLAAS-3 cloud detection scores as a function of the minimum COT threshold used to discriminate between clear and cloudy CALIOP observations. POD is the Probability Of Detection, FAR is the False Alarm Ratio and KSS denotes the Hanssen-Kuipers Skill Score.**

For the comparisons with observations from SYNOP stations, the time span from 2004-2020 is validated, after applying restrictions to the SYNOP data used, based on the criteria described in Section 3.2. Figure 4a shows the mean CFC difference between CLAAS-3 and SYNOP at all selected SYNOP sites, averaged for the period 2004-2020. CLAAS-3 CFC is larger than SYNOP in the Middle East and smaller over the Iberian Peninsula, northern Africa and at coastal sites in South America. Coastal SYNOP sites generally pose an extra challenge in the CLAAS-3 evaluation: the matched CLAAS-3 grid cells can be classified as water, whereas the SYNOP site location is on land. There are also SYNOP-related inaccuracies in CFC observations: overestimation of cloudiness due to the obscuring of cloud-free spaces by convective clouds with high vertical extent (e.g. Karlsson, 2003) and underestimation during night because of difficulties in observing semi-transparent



cirrus clouds. However, it is difficult to quantify the effects of these inaccuracies here, due to the extensive temporal averaging.

Averaging results from all SYNOP sites and corresponding CLAAS-3 averages on a monthly basis leads to the time series results shown in Figure 4b. The results show an overall good agreement, with monthly biases always lying between ±5 %. The apparent seasonality in both data sets, with maximum cloud cover during northern hemisphere winters, is due to the fact that most SYNOP sites are located in the northern hemisphere. Hence, the time series pattern reflects the seasonal cycle of cloudiness over those latitudes.





Figure 4: (a) Mean CFC difference between CLAAS-3 and SYNOP cloud cover at each preselected SYNOP site for the period 2004-2020. (b) Time series of monthly mean and annual mean (lines with symbols) CFC from CLAAS-3 and SYNOP. The values represent averages calculated from all stations shown on the map.



Comparison with CALIOP offers the possibility to evaluate CLAAS-3 cloud fraction for three vertical layers separately. As explained in Section 2.2.1, high, middle and low cloud layers are defined based on CTP thresholds. Figure 5 shows the resulting time series of CFC for high-, mid- and low-level clouds as well as the total cloud fraction from CLAAS-3 and the two CALIPSO-GEWEX flavors. Results span the period from June 2006 to December 2016, when the two data sets overlap, and the averages were computed after excluding regions where the SEVIRI VZA exceeds 75°. The differences between

passive and top-layer flavors are small for low and middle clouds (~2%), but more prominent for high clouds. In fact, the filtering of very thin high clouds in the passive flavor data set leads to CLAAS-3 low- and mid-level cloud CFC being higher than the passive flavor. For total and high-level CFC, CLAAS-3 lies between the two flavors. This shows that while CLAAS-3 does not contain all optically thin high clouds detected by CALIOP, it contains a considerable fraction of those with COT less than 0.3 (the threshold applied in the passive flavor, see Sect. 3.1), as was also inferred by the detection

scores analysis (Figure 3). In the middle cloud layer, CLAAS-3 has higher CFC than both flavors, while results for low clouds are more mixed.

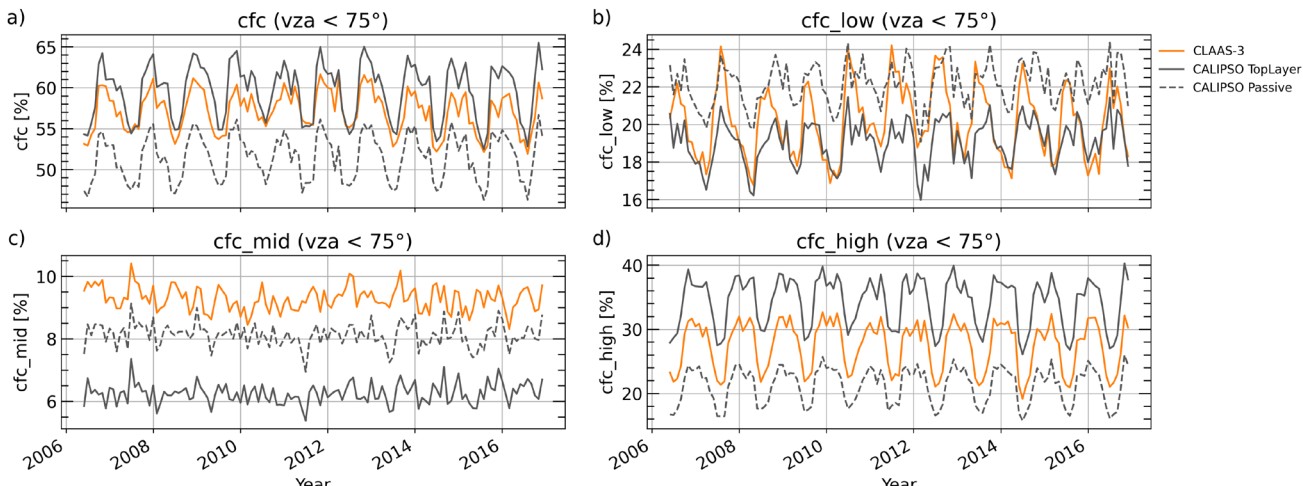

**Figure 5: Time series of spatially averaged monthly mean CFC from CLAAS-3 and CALIPSO-GEWEX top layer and passive**
**flavors for total CFC (a) low- (b) mid- (c) and high- (d) level clouds.**

### 4.2 Cloud phase

As in the CMa case, level 2 CPH data were averaged in a regular 1.5° × 1.5° grid and compared with corresponding collocated data from CALIOP. Based on the cloud detection score analysis (Figure 3), the uppermost cloud layers from the CALIOP profiles with ICOT (top-down integrated COT) = 0.1 are excluded from this comparison. Results are shown in

Figure 6. Overall patterns are similar in both data sets. Liquid cloud fraction in CLAAS-3 is higher in higher latitudes, locally up to 30%, due to greater amounts of optically thin ice clouds detected by CALIOP, which reduce the fraction of


liquid clouds. Negative differences appear around the ITCZ, . This is because ice clouds (high convective and outflow cirrus) dominate there, and the CALIOP top layer filtering results in an increased CALIOP liquid cloud fraction compared to CLAAS-3 (i.e. mainly ice clouds are excluded). Positive differences over desert areas are also apparent, but they come from

a low number of cases, since the presence of clouds is scarce over these areas.

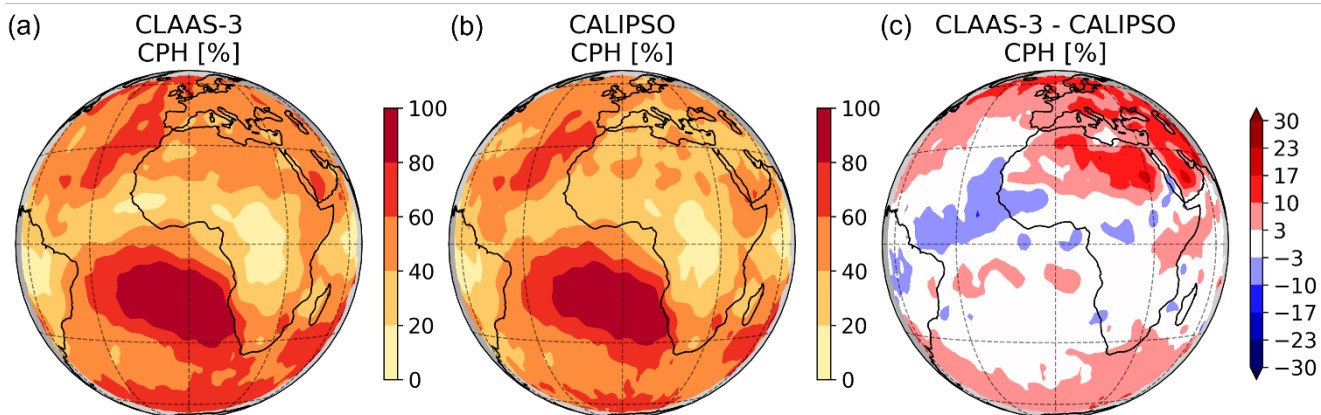

**Figure 6: Spatial distribution of the 2013 average level 2 liquid cloud fraction (CPH) from CLAAS-3 (a), CALIOP (b), and their difference (c). Here CPH is estimated by collecting matchups to a regular 1.5° × 1.5° grid and averaging them within each grid**

**box. The CALIOP CPH is taken from the layer where the ICOT exceeds 0.1. A 2-dimensional Gaussian filtering was used for noise reduction.**

Figure 7 shows how the liquid cloud fraction detection scores change when we vary the uppermost CALIOP ICOT that is excluded from the comparisons. As this value increases, many CALIOP profiles switch from ice to liquid phase. The fact that the liquid POD decreases, suggests that many of the excluded thin ice cloud cases were correctly retrieved by CLAAS-3.

The rapid decrease in liquid FAR can be explained by the decrease in cases where CALIOP detected ice clouds, whereas CLAAS-3 retrieved liquid clouds. Similarly to the CMa case, KSS peaks at ICOT values between 0.1 and 0.2, suggesting that CLAAS-3 CPH provides the best separation of liquid and ice cases at this optical thickness threshold.

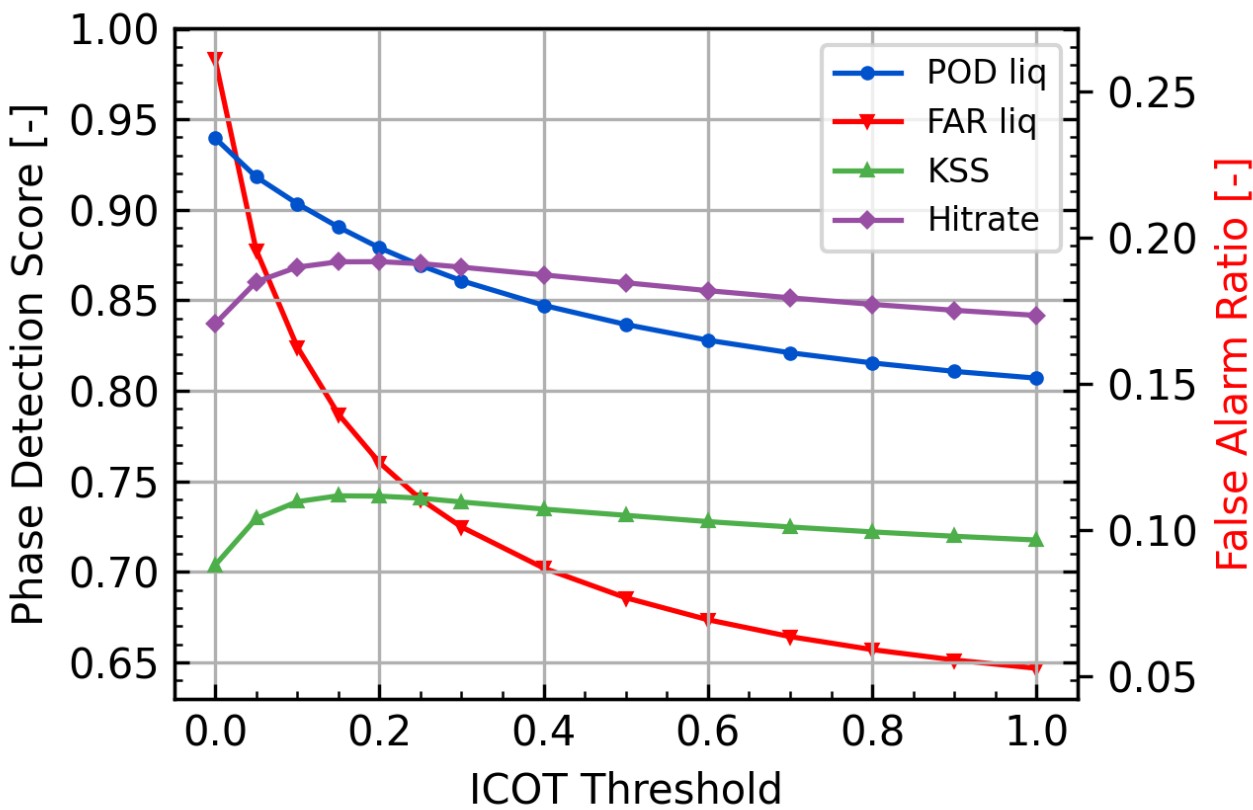

**Figure 7: CLAAS-3 cloud phase detection scores as a function of the ICOT threshold, which determines the reference (uppermost) CALIOP cloud layer. POD is the Probability Of Detection and KSS denotes the Hanssen-Kuipers Skill Score.**

### 4.3 Cloud-top height

The level 2 CTH was also compared with CALIOP collocations aggregated and averaged in a regular 1.5° × 1.5° grid, as in the CMa and CPH cases. The results are shown in Figure 8. As in the CPH case, the CALIOP CTH comes from the uppermost layer after excluding the top layers where ICOT = 0.1. CTH large-scale spatial patterns compare well. CALIOP has overall higher cloud top height, with absolute differences below 1 km in most cases.

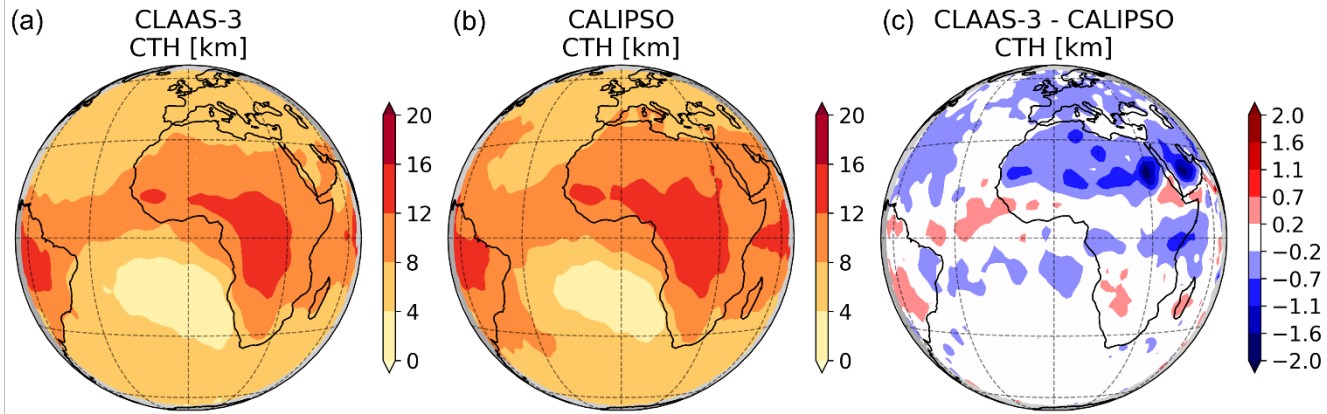

**Figure 8: Spatial distribution of the 2013 average level 2 Cloud Top Height (CTH) from CLAAS-3 (a), CALIOP (b), and their difference (c). Here CTH is estimated by collecting matchups to a regular 1.5° × 1.5° grid and averaging them within each grid box. The CALIOP CTH is taken from the layer where the top-down integrated COT (ICOT) exceeds 0.1. A 2-dimensional Gaussian filtering was used for noise reduction.**

In the case of level 3 products, CTH is compared with CALIPSO-GEWEX (both passive and top layer flavors) and MODIS. Figure 9 shows the time series of averaged values from all data sets, with the requirement that they lie in the region where SEVIRI VZA is less than 75°. All time series included in Figure 9 are stable and show a very similar seasonal variability. The CLAAS-3 results lie between the top layer and the passive flavor from CALIPSO-GEWEX, verifying its ability to detect part of the high thin clouds with COT lower than 0.3. In terms of absolute differences, CLAAS-3 lies slightly closer to the CALIPSO-GEWEX passive flavor than the top layer one. On the other hand, the differences between CLAAS-3 and MODIS CTH typically exceed 2000 m, probably due to the different retrieval approaches used: MODIS uses the $CO_2$ slicing technique (Baum et al., 2012), whereas CLAAS-3 uses a neural network approach with a training data set compiled from CALIOP data collocated with SEVIRI. The latter approach also explains the better performance of CLAAS-3 compared to MODIS (considering CALIPSO-GEWEX as a reference), which was also discussed in Håkansson et al. (2018), where a similar approach was applied to MODIS data.





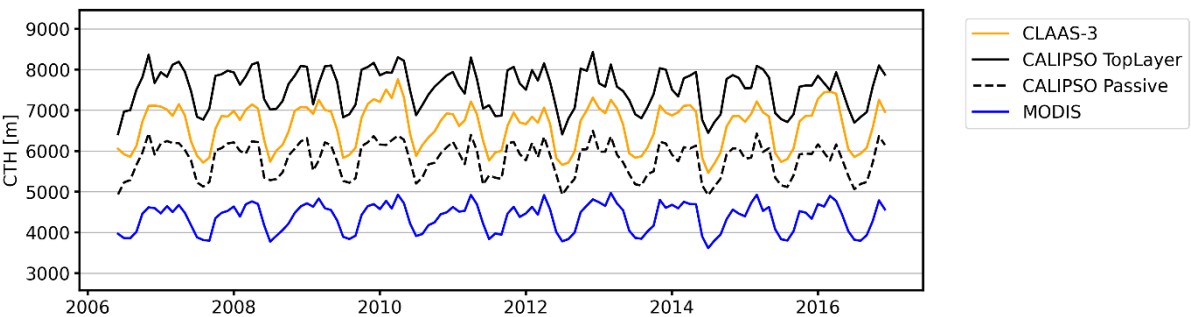

**Figure 9: Time series of spatially averaged monthly mean CTH from CLAAS-3, MODIS and CALIPSO-GEWEX top layer and passive flavors. The spatial averaging was performed over the region where the SEVIRI Viewing Zenith Angle is lower than 75°.**


## 4.4     Liquid CPP products

Here we compare level 2 liquid CPP products with AMSR2 (LWP, LWP error), MODIS (COT, CRE, CDNC, CGT) and in situ measurements (CDNC), and level 3 all-sky LWP with MAC-LWP.

A correlation coefficient equal to 0.79 was estimated based on all valid collocations of CLAAS-3 level 2 LWP with AMSR2,
showing that the two data sets agree well (Figure 10a). Their bias is also small (0.11 g m$^{-2}$), while the standard deviation of the difference is close to 50 g m$^{-2}$, revealing a considerable scatter.

Using AMSR2 data as reference, the CLAAS-3 LWP uncertainty can also be evaluated, by comparing it with the absolute difference between CLAAS-3 and AMSR2 LWP. Such an uncertainty analysis neglects the following. Firstly, uncertainty estimates for AMSR2 LWP were not available and thus not included. Secondly, CLAAS LWP uncertainty estimates do not
include errors caused by violation of the assumptions of horizontal and vertical homogeneity. Finally, remaining uncertainties from colocation and/or representativity are assumed to be small. As the balance between these constraints is not known a sound conclusion on the adequacy of CLAAS-3 uncertainty estimates is not possible. Nonetheless, the following is observed, assuming that the uncertainty from CLAAS-3 is the dominant uncertainty source: as the validity of the absolute bias being smaller than the (total) uncertainty is valid statistically only, a reasonable spread around the 1:1 line is expected
and actually observed, except for a region that covers the full range of differences at small CLAAS-3 uncertainty estimates. The correlation found (r=0.61, Figure 10b) points to a reasonable validity of the CLAAS-3 algorithm in estimating LWP uncertainty.

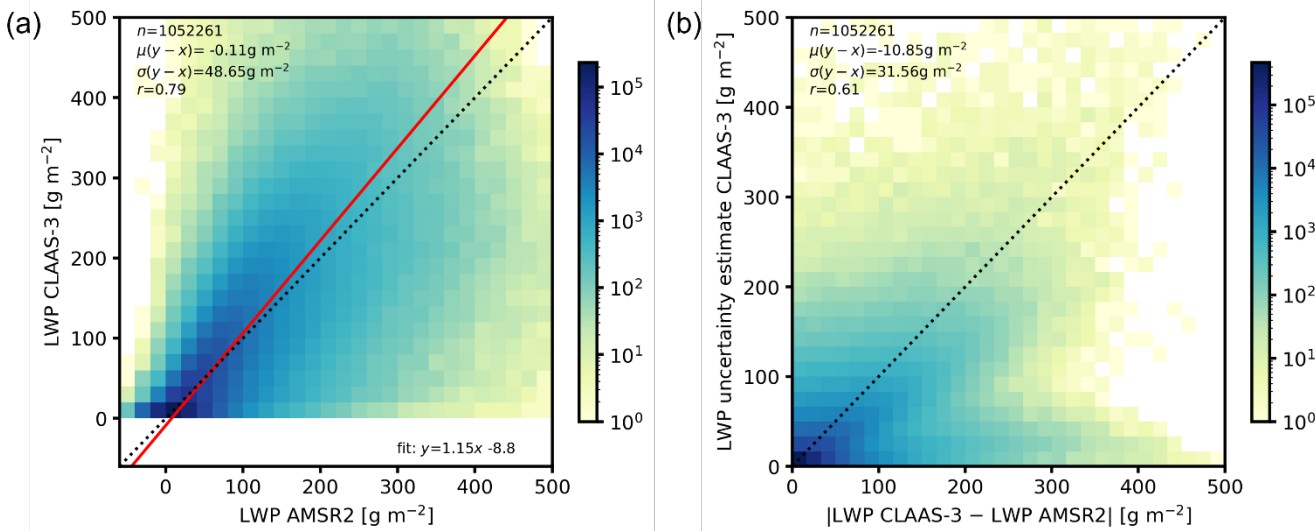

**Figure 10: Comparisons of CLAAS-3 level 2 LWP (retrieved using the 3.9 μm channel) with AMSR2. a) Scatter plot and statistical scores CLAAS-3 vs. AMSR2 LWP. b) Scatter plot of CLAAS-3 LWP uncertainty vs. the absolute difference of CLAAS-3 and AMSR2 LWP values. All collocations (available only over ocean) in March 2013 were used. Statistics including the number of collocations, the mean difference, the standard deviation of the difference and the linear correlation coefficient are indicated. The red line in panel (a) shows an orthogonal fit to the data.**

As with comparisons against AMSR2, CLAAS-3 level 2 liquid COT, CRE, CDNC and CGT were compared against corresponding MODIS data, following the collocation methodology described in Section 3.3.

Comparison results from the 3.9 μm (3.7 μm for MODIS) retrievals are shown in Figure 11. The mean difference in COT is close to 2, with CLAAS-3 being on average lower than MODIS. The liquid CRE comparison (Figure 11b) shows a similar correlation with CLAAS-3 values as in the liquid COT case. Dependencies of differences on SZA and VZA were investigated. This analysis indicated that the COT difference primarily depends on SZA, with CLAAS-3 showing a less strong increase towards high solar zenith angles than MODIS, and the CRE difference primarily depends on the SEVIRI VZA, with CLAAS-3 showing lower values than MODIS at low VZA and higher values towards the edge of the disk (not shown).

As mentioned in Section 3.3, CDNC and CGT are not included in MODIS level 2. Instead, they were computed based on the same relations as in CLAAS-3 (Bennartz and Rausch, 2017). The resulting comparisons are shown in Figure 11c and d. The average CDNC difference between the two data sets is negligible, but the large standard deviation suggests relevant discrepancies in spatial features. Indeed, CLAAS-3 – MODIS CDNC differences showed a comparable dependency on the SEVIRI VZA as CRE but with an opposite sign. Finally, CGT shows again a good correlation with MODIS and a modest negative mean difference.

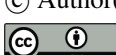

**Figure 11:** Scatter plots of liquid CPP level 2 products with MODIS Terra and Aqua data as reference: (a) COT, (b) CRE, (c) CDNC, (d) CGT at 3.9 micron. . Statistics are indicated in the plots as outlined in the caption of Figure 10. Note that the number of collocations is higher for COT than for the other variables because for a portion of the cloudy pixels no solution for CRE, and thus CDNC and CGT, can be found.

In addition to the level 2 evaluation of liquid cloud properties with similar satellite-based retrievals, CDNC was also compared with in situ measurements from the ORACLES project flights. The relevant collocation process and quality criteria applied are described in Section 3.5.

The resulting comparison outcome is shown in Figure 12. The correlation is reasonable, considering the radically different approaches used in producing these two data sets. The bias shows that CLAAS-3 CDNC is on average lower than the in situ





measurements. While causation cannot be established from this analysis, this is an effect that should be expected in the presence of absorbing aerosols above clouds: the aerosols would cause a reduction in the reflectances measured by the satellite, which in turn would lead to retrievals of higher CRE and lower COT, and thus lower CDNC. In fact, in most of the

85 cases with AAI>2 that were excluded from this analysis, CLAAS-3 CDNC was considerably lower than the measurements.

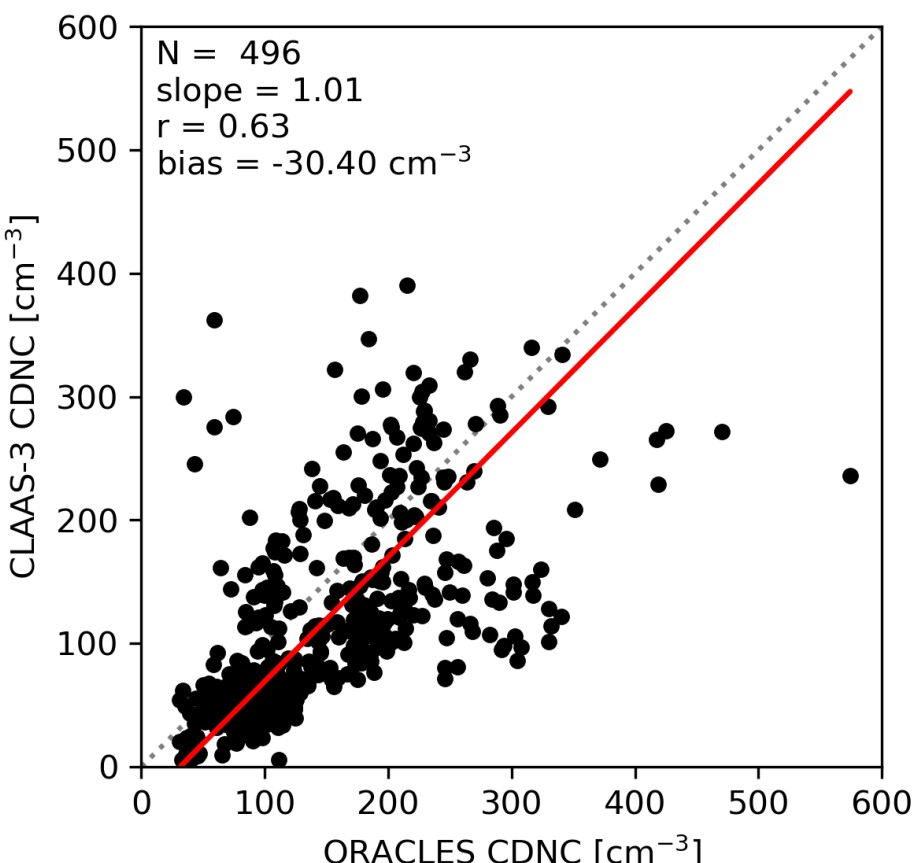

**Figure 12: Scatter plot and statistical scores of CLAAS-3 level 2 CDNC vs. measurements from the ORACLES campaign flights,**
**averaged in each CLAAS-3 grid cell. The dotted line is the 1:1 line and the red line is the one resulting from the linear regression.**

The level 3 LWP was compared against MW-based retrievals from the MAC-LWP data set. Both monthly average and monthly mean diurnal values were assessed, focusing on the marine stratocumulus region of the southeastern Atlantic, as explained in Section 3.4.

Figure 13a shows the resulting time series of CLAAS-3 and MAC-LWP monthly mean all-sky LWP, averaged for time slots between 07:00 and 15:00 UTC, when both data sets cover the region adequately. They agree well in terms of seasonality and





have a similar range of values. CLAAS-3 appears slightly higher from August to November and lower from January to April.

The diurnal variation in all-sky LWP over the same region is shown in Figure 13b. Values are calculated (per time slot) as 555 averages of the entire available period (2004-2016). Both data sets show a decrease in the all-sky LWP, which is expected due to clouds breaking up during the day. CLAAS-3 exhibits a somewhat stronger diurnal cycle, with higher values at the beginning and end of the day, coinciding with (and possibly caused by) high solar zenith angles.

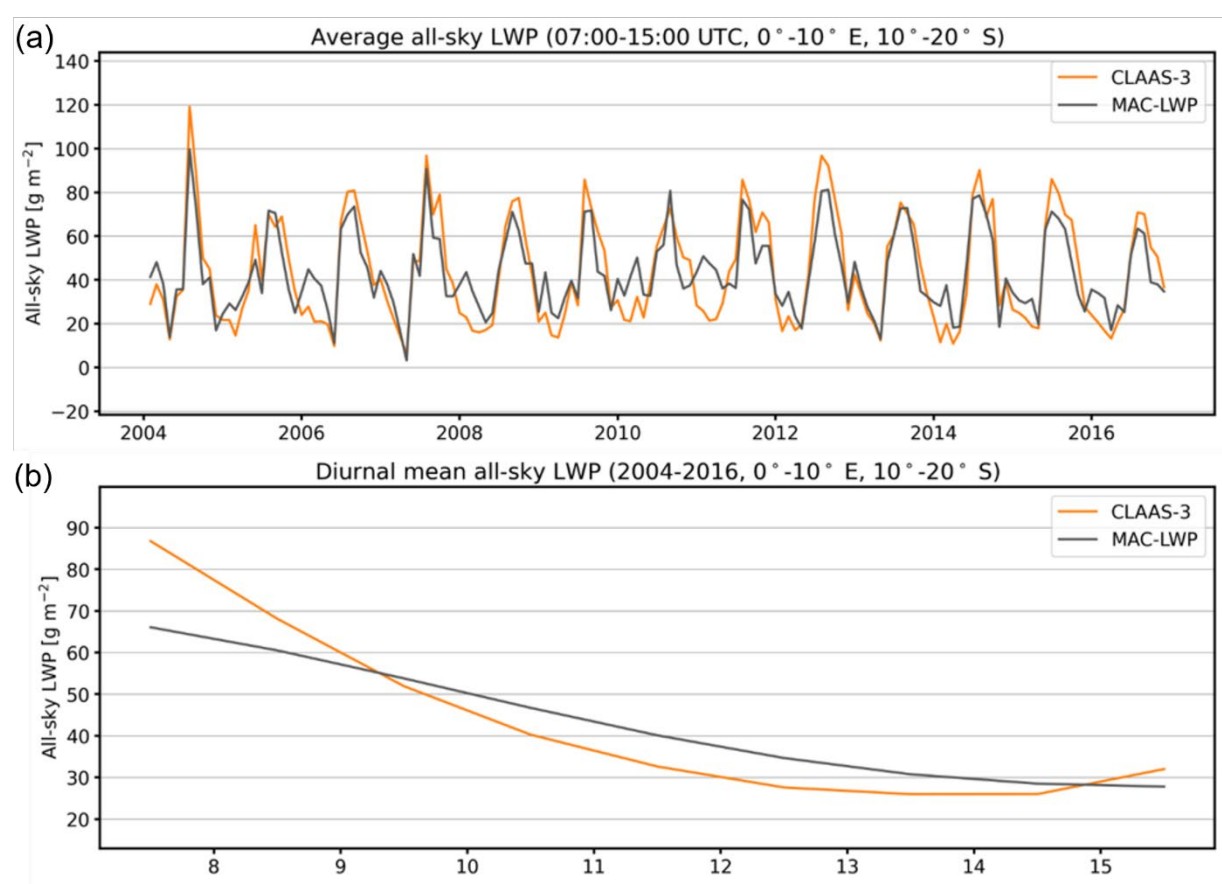

**Figure 13: (a) Time series of monthly mean all-sky LWP from CLAAS-3 and MAC-LWP. Values are spatial averages from the region 10°-20° S, 0°-10° E, and temporal averages of time slots between 07:00 UTC and 15:00 UTC. (b) Monthly mean diurnal variation of all-sky LWP from CLAAS-3 and MODIS over the same region, calculated from monthly averages in the period 2004-2016.**



## 4.5    Ice CPP products

Figure 14 shows comparisons of CLAAS-3 level 2 ice cloud properties with DARDAR, based on the collocation data set described in Section 3.5. Ice COT (Figure 14a) agrees reasonably well, with a small average difference. However, DARDAR COT has a larger dynamical range, including more thin (COT < 1) as well as more thick (COT > 30) clouds. Comparison results for ice CRE derived using the 3.9 μm channel, however, show practically no correlation with DARDAR (r=0.12), while a considerable bias of -19 μm is also found. This bias becomes even much larger (-29 μm), if the DARDAR CRE is calculated by applying a vertically uniform weighting, instead of the weighting used in Figure 14b, which emphasizes the uppermost part of the cloud (Section 3.5). The 1.6 μm-based CRE retrievals are on average almost 10 μm larger than those based on 3.9 μm, (not shown here), which is partly related to radiation at 1.6 μm penetrating deeper into the cloud, where ice crystals are normally larger. Consequently, the 1.6 μm-based CRE agrees somewhat better with DARDAR in terms of bias, with also a higher correlation (r=0.37). Similar large differences and weak correlations between passive visible–near-infrared and active lidar-radar retrievals of CRE have been found before (Stein et al., 2011), and are still not fully understood.

When comparing with MODIS level 2 data, the ice COT values show no systematic difference between the two data sets (Figure 14c), with the standard deviation, however, being somewhat larger than in the liquid clouds case. The CLAAS-3 and MODIS level 2 ice CRE show good agreement (Figure 14d), with a bias of -3.5 μm. This level of agreement could be anticipated, considering that the same ice cloud model is used in both retrievals (Section 2.3).

**Figure 14: Comparisons of level 2 ice COT and CRE with DARDAR (panels a and b) and MODIS Terra and Aqua (panels c and d). CLAAS-3 data are from the 3.9 μm retrievals. The DARDAR CRE was weighted towards the top of the cloud ($\tau_w = 1$ in Eq. 1) to reflect the vertical sensitivity of the CLAAS-3 CRE retrievals. All collocations are from March 2013. Note the logarithmic scale in panel (a). Statistics are indicated in the plots as outlined in the caption of Figure 10.**

Figure 15 shows time series of monthly spatial averages of IWP, ice COT and CRE in the region 45° S-N, W-E from CLAAS-3 and MODIS. Comparison of the IWP shows excellent agreement in terms of seasonal patterns although there is a bias of around 50 g m$^{-2}$, with MODIS being consistently larger. This can be attributed to both ice COT and CRE differences, with CLAAS-3 being lower than MODIS by 1.0 and 6.3 μm on average, respectively. Nevertheless, in both cases seasonal

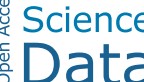

characteristics agree well. Note that monthly mean COT and CRE biases differ from the level 2 results shown in Figure 14c and d. This can be explained by the different spatial coverage (45° S-N and W-E for level 3 versus full disk for level 2) and CLAAS-3 temporal sampling (full day for level 3 versus MODIS overpass times for level 2). The main objective of Figure 595 15 is to evaluate the consistency in seasonal variability and long-term trends rather than mean values.

**Figure 15: Time series of monthly mean (in-cloud) IWP (a), ice COT (b) and ice CRE (c) from CLAAS-3 and MODIS. All values are spatial averages of common occurrences in the region 45° S-N and W-E.**





## 5    Data availability

The CLAAS-3 data record DOI is 10.5676/EUM_SAF_CM/CLAAS/V003 (Meirink et al., 2022). All intellectual property rights of the CM SAF CLAAS-3 products belong to EUMETSAT. The use of these products is granted to every interested user, free of charge. If you wish to use these products, EUMETSAT's copyright credit must be shown by displaying the words "copyright (year) EUMETSAT" on each of the products used. The data can be ordered via

https://doi.org/10.5676/EUM_SAF_CM/CLAAS/V003. The same link contains relevant documentation: a product user manual, validation report and algorithm theoretical basis documents, and related publications. Links to auxiliary data, tools and previous CLAAS versions can also be found there and via the CM SAF website https://www.cmsaf.eu.

## 6    Conclusions

This study provided an overview of CLAAS-3 and its evaluation. CLAAS-3 is a climate data record of cloud properties

produced by the EUMETSAT CM SAF, based on measurements from SEVIRI sensors on board geostationary satellites Meteosat-8, 9, 10 and 11. When used in CLAAS-3, all these satellites are at (or close to) the 0.0° longitude sub-satellite point, offering good coverage of Africa, Europe and the Atlantic Ocean, starting in February 2004 and extending to the present. CLAAS-3 provides a large amount of cloud variables, related to their height, phase, optical and microphysical characteristics.

All CLAAS-3 properties were evaluated based on independent data sets which were considered as reference. The majority of these comes from satellite-based retrievals (CALIPSO, MODIS, MW imagers, DARDAR), but ground-based observations (SYNOP) and in situ measurements (ORACLES) were also used. In summary, results show that CLAAS-3 agrees overall well with satellite-based reference data from similar (e.g. MODIS) or different sensors and retrieval approaches (e.g. lidar, radar and MW observations). Comparison results are also promising in the cases of completely unrelated observation

approaches (i.e. ground-based and in situ measurements). Observed discrepancies can most of the times be traced back to known issues in similar evaluation attempts: different retrieval approaches, different underlying assumptions in otherwise similar retrieval methods, and different sampling conditions (temporal, spatial, illumination) of the two data sets under comparison.

The range of CLAAS-3 processing levels (from instantaneous to monthly averages), and the 19-year long and continuously

extended time series, offer a similarly large range of possible uses: from local to continental scales, and from minute-scale processes to long term tendencies. Considering the extent of usage of the previous CLAAS versions, discussed in the Introduction, a promising potential for CLAAS-3 becomes apparent: the addition of new variables and the extension of the temporal coverage suggest that CLAAS-3 can be useful for an even larger number of users and applications.

Future continuation of CLAAS data records production is also secured by CM SAF. Presently, preparations are under way

for the production, in a few years, of the fourth edition of CLAAS. Apart from potential improvements in SEVIRI calibration and retrieval algorithms, in CLAAS-4 it will be attempted to include the Flexible Combined Imager (FCI) on board the



Meteosat Third Generation Imaging (MTG-I) satellites in a seamless way; FCI will provide observations at higher spatial and temporal resolution and in slightly different channels, including several new, extending and improving SEVIRI-based data records for more than 20 years.

## 7    Author contributions

NB wrote the first version of the paper and performed part of the cloud physical properties evaluation. IS and MSt developed and implemented algorithms for level 3 calculations, and performed the cloud mask, fraction and cloud-top evaluation. IH was responsible for the operational production of the data record. KGK, NH, EJ and SE developed and validated the algorithms for cloud detection and cloud-top properties retrieval. MSc and RH provided input on the structure and contents of the manuscript. JFM developed and implemented the cloud physical properties algorithm, and performed part of the evaluation of corresponding products. All authors reviewed and edited the manuscript.

## 8    Competing interests

The authors declare that they have no conflict of interest.

## 9    Acknowledgements

This work was performed within the EUMETSAT CM SAF framework and all authors acknowledge the financial support of the EUMETSAT member states.

CALIOP data were obtained from the NASA Langley Research Center Atmospheric Science Data Center.

We thank DWD for providing data from SYNOP stations.

MODIS data were acquired from the Level-1 and Atmosphere Archive & Distribution System (LAADS) Distributed Active Archive Center (DAAC), located in the Goddard Space Flight Center in Greenbelt, Maryland (https://ladsweb.nascom.nasa.gov/).

AMSR data are produced by Remote Sensing Systems and were sponsored by the NASA AMSR-E Science Team and the NASA Earth Science MEaSUREs Program. Data are available at www.remss.com.

MAC LWP data were obtained from the Goddard Earth Science Data and Information Services Center (http://disc.sci.gsfc.nasa.gov).

All ORACLES in situ data used in this study are publicly available at https://espo.nasa.gov/oracles/archive/browse/oracles/P3/PDI-CDNC.

We thank the AERIS/ICARE Data and Services Center (http://www.icare.univ-lille1.fr/) for providing access to the DARDAR data used in this study.



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
