# Peer review of "CLAAS-3: the third edition of the CM SAF cloud data record based on SEVIRI observations"

_Earth System Science Data, 2023_

## Referee Comment (RC2)

Reviewer comments for "CLAAS-3: the third edition of the CM SAF cloud data record based on SEVIRI observations" by Nikos Benas, et al.

General impression: Major Revisions.

General Comments: I get a repeated impression that very old data was used for comparisons, especially for MODIS. Since this entire paper is about comparing data records, getting some serious clarification about the MODIS data sources, especially those used in images, is absolutely in order. Studies that were based on data as old as MODIS Collection 4, created before Aqua was even completed, let alone launched, are quoted in the comparisons. MODIS Collection 4 suffered from some severe deficiencies, especially for 3.7um retrieval. Things got a bit better for C5 and finally mostly remedied in C6.1. That'd be about all there is to say at this point.

Specific Comments:

Figure 1: maybe add some clarifying text as to the time periods the backups were used. I believe the thin data strips are the backup periods for the ones with the more continuous lines, but it might not be clear to the user. Also what are the implications of using backups? For example, I imagine Meteosat-8 and Meteosat-9 would show some deterioration in data after that many years.

Line 90-95: What about the thermal channels? Was the calibration slope determined for both main and backups?

Line 110: what do you do with the data that falls between 75 and 95 degrees SZA?

Line 143: so does this new cloud mask algorithm actually perform better for marine stratus off Africa at sunrise and sunset? Or the issue is still there and that is why you're cutting out your solar zenith that way?

Line 151: Your training dataset is based on CALIOP, which means *all* your training data is early afternoon. Yet you apply the algorithm to all observation times. How does that affect the accuracy of your neural network? What are your uncertainties outside of 1:30pm local time? You say that you calculate them, but what kind of value ranges are you getting?

Line 175: Your cloud microphysics retrievals methods are eerily similar to SEV06-CLD. I am wondering why that product is not mentioned in any way.

Line 194: replace "… on board the Cloud-Aerosol Lidar and Infrared Pathfinder Satellite Observation (CALIPSO) satellite" with "on board the Cloud-Aerosol Lidar and Infrared Pathfinder Satellite Observation (CALIPSO) spacecraft". Repetitive.

Line 220: could you just drop some ballpark uncertainty values that you encounter with CALIOP? For more discussion, of course the reference, but a few numbers here would do some good.

Line 260: If you would've looked at SEV06-CLD product that is the MODIS C6.1 code running on SEVIRI, you would have an exact uncertainty comparison due to instrument differences rather than waving of hands you are doing here.

Line 320: CER actually is a rather mixed bag, heavily influenced by above-cloud aerosol. You can't really make a statement that cloud drops get larger towards the cloud top. Bennartz and Rausch are careful to indicate that it's not always true. Moreover if there is any ice cloud in the scene at all, 3.7um will always be smaller than 1.6um. Anyways, it's more complicated. I don't know what impact if any that has on your conclusions, but still.

Line 335: Again, that is a very old study that was moreover influenced by outright bugs in the 3.7um retrieval algorithm at the time. The issues were not corrected until Data Collection 5 was released. Please don't quote that paper. What impact this has on your use of DARDAR data, I couldn't tell you, but I would not base a single thing on that paper, particularly because you're making statements about penetration depth and retrieval values. Retrieval values were garbage, differences in penetration depth completely aside.

Line 385: Can you show an image same as figure 2, that the bias actually decreases with additional screening criteria applied? That -30%. Oooof.

Figure 4: It seems from this image that CLAAS-3 is picking up persistent aerosol loads as clouds. It is very dusty in the areas with the highest positive biases. Can you explain?

Figure 11: Don't tell me you used Data Collection 5 MODIS data! For Collection 6.1 each retrieval pair has its own optical thickness reported. This kind of difference in count between tau and re was only present in C5 data and older. Please explain.

Figure 14: Same thing. What data collection did you use here? This doesn't look right at all, if you used C6.1. Number of successful retrievals for optical thickness and effective radius is identical and each band combination has its own tau, re and water path. They've been entirely decoupled in the latest MODIS data collection.

---

## Author Comment (AC1)

*General Comments*

*The paper summarises the new reprocessing of the SEVIRI satellite cloud property data set. The data set has been extended in time, both forwards and backwards and additional parameters have been added such as cloud geometrical thickness and cloud droplet number concentration. The quality of the data set is high and it is clear a lot of work has gone into assessing the underly FCDR as well as the L2 and L3 products. The data set will be useful for many different weather and climate studies.*

*The paper is very thorough, products have been validated and compared at level2 (swath) and at level3 (monthly).*

*The products have been evaluated with the most appropriate reference data sets and with aircraft flight campaign data. It was very pleasing to see discussion of uncertainties on the data set and even evaluation of the uncertainty of the LWP data set, this is in line with best practice.*

*The paper is very clearly written and I found very few technical errors*

We are grateful to Caroline Poulsen for dedicating time to deliver this positive review. Below we provide answers to her comments, and address her remarks aiming to improve the quality of the manuscript accordingly.

*Specific Comments*

*For the comparison of CTH with Calipso it would have been good to see a plot of the standard deviation (Figure 8) and/or some discussion on what compensating errors there could be for example boundary layer clouds could be too high (miss classification of inversions) while cirrus clouds too low.*

Thank you for this suggestion. Indeed, Figure 8 alone is not informative regarding the variability in CTH. In the revised manuscript we will add maps in Figure 8 showing the standard deviations in CTH separately from CLAAS-3 and CALIOP, and the standard deviation of their difference. The revised figure is also shown below. The standard deviations of the two data sets are in good agreement, with lower values occurring in stratocumulus clouds (southeastern Atlantic) and higher values in part of the ITCZ, possibly associated with strong convection and frequent cirrus outflow. The standard deviation of the differences ranges between 1 and 4 km, with higher values also occurring near the equator.

[Figure]

[Figure]

**Figure 1: Spatial distribution of the 2013 average level 2 Cloud Top Height (CTH) from CLAAS-3 (a), CALIOP (b), and their difference (c). Here CTH is estimated by collecting matchups to a regular 1.5° × 1.5° grid and averaging them within each grid box. The CALIOP CTH is taken from the layer where the top-down integrated COT (ICOT) exceeds 0.1. A 2-dimensional Gaussian filtering was used for noise reduction. Standard deviations of CLAAS-3 CTH, CALIOP CTH and of their difference are shown in (d), (e) and (f), respectively.**

Regarding possible compensating errors that could erroneously lead to a good agreement in the average values, we have plotted histograms of the occurrence of CTH from CLAAS-3 level 2 and CALIOP, separately for liquid and ice clouds, which show that the occurrence of such errors is not likely. The histograms are shown below, with similar ones discussed also in the CLAAS-3 validation report (CM SAF 2022d).

[Figure]

**Histogram of occurrence of level 2 cloud top heights from CLAAS-3 and CALIOP, separately for liquid and ice clouds. Dotted lines show cases when the CALIOP CTH is taken from the layer where the top-down integrated COT exceeds 0.1.**

As in Figure 8, the CALIOP CTH is taken from the layer where the top-down integrated COT (ICOT) exceeds 0.1. Results show that the agreement is overall good, especially for the liquid phase. It also appears that CLAAS-3 misclassifies a small amount of ice mid-level clouds as liquid (at CTHs of 5-6 km), while it contains more high ice clouds than CALIOP (at CTH near 14 km).

*Figure 10 why are there negative values of LWP AMSR2?*

Negative LWP values result from the retrieval algorithm and are not anymore forced to zero in the RSS products since version 7. We will clarify this by adding the following sentence after L491: "Note that the AMSR2 LWP retrievals from version V8.2 include negative values, which arise as a consequence of random errors in brightness temperatures, and are not forced to zero since that would lead to a positive LWP bias (Elsaesser et al., 2017)."

*Section 4.5 have the authors considered that the difference between CLASS and MODIS could also be due to different liquid/ice cloud fractions?*

Indeed, this is a possibility that was not discussed in the manuscript. While we compare in-cloud only IWP, differences due to different cloud phase may arise when temporally averaging the instantaneous/daily values to monthly mean values. We will discuss this point in the revised manuscript. Hence, a possible explanation of the systematically higher MODIS IWP would be if MODIS

detects less (high) thin ice clouds than CLAAS-3. While we have not investigated this further, the systematically higher CTH in CLAAS-3 compared to MODIS, shown in Figure 9, supports this hypothesis.

*Technical errors*

*line 337 insert new line*

Done.

*while it maybe obvious that the x-axis in plots is year it would still be good to have this labelled.*

Labels added in Figs. 4, 9, 13 and 15.

*x-axis labelling on Figure 13*

Added.

---

## Author Comment (AC2)

*General Comments: I get a repeated impression that very old data was used for comparisons, especially for MODIS. Since this entire paper is about comparing data records, getting some serious clarification about the MODIS data sources, especially those used in images, is absolutely in order. Studies that were based on data as old as MODIS Collection 4, created before Aqua was even completed, let alone launched, are quoted in the comparisons. MODIS Collection 4 suffered from some severe deficiencies, especially for 3.7um retrieval. Things got a bit better for C5 and finally mostly remedied in C6.1. That'd be about all there is to say at this point.*

We would like to thank Anonymous Referee #2 for taking the time to deliver this constructive review. Below we provide our point-by-point replies, along with plans to revise the manuscript accordingly, aiming to improve its clarity and quality.

*Specific Comments:*

*Figure 1: maybe add some clarifying text as to the time periods the backups were used. I believe the thin data strips are the backup periods for the ones with the more continuous lines, but it might not be clear to the user. Also what are the implications of using backups? For example, I imagine Meteosat-8 and Meteosat-9 would show some deterioration in data after that many years.*

Specific dates when backup sensors were used are given in table 3-4 of the CLAAS-3 ATBD (CM SAF 2022a). We will include the relevant text and reference in the revised manuscript. We will also update the caption of Figure 1 to clarify that the thin data strips are backup periods.

Regarding possible implications of using backups, no apparent issue appears in level 3 monthly mean data (note that the maximum number of days for which backup data was processed in any one month is 9, which occurred in January 2019). In level 2, we examined a specific case in November 2015, when the prime satellite MSG-3 was temporarily replaced by MSG-1. Below we have plotted time series of retrieved variables averaged over a large (2000x2000 pixels centered around 0° longitude-latitude) area every 15 minutes. The time series include the switch from MSG-1 to MSG-3 on 18 November 2015 between 11:45 and 12:00 UTC. The colored lines for the two satellites are interconnected with a solid black line, which helps to judge the continuity of the transition. The figures suggest that for all cloud properties the transition between satellites is smooth. Similar findings were obtained for other transitions. Note that for variables that are retrieved only during day (COT, CRE, CWP, CDNC) an additional criterion of the solar zenith angle being lower than 60° was imposed, and that the fraction of valid retrievals in the area was required to be 90% relative to noon (when the whole area is in daylight).

[Figure]

*Line 90-95: What about the thermal channels? Was the calibration slope determined for both main and backups?*

Yes, the EUMETSAT operational calibration slopes for thermal channels are available separately for each sensor. This will be clarified in the revised manuscript. When processing backup sensors, calibration slopes are switched automatically.

*Line 110: what do you do with the data that falls between 75 and 95 degrees SZA?*

The way this sentence was phrased is not clear. The mentioned SZA thresholds refer to the selection of level 2 data for the separation of level 3 monthly data in day and night products, rather than a general "day and night definition in CLAAS-3". In the revised manuscript, we will rephrase this sentence to avoid any misunderstanding that data gaps occur between 75 and 95 degrees SZA. Instead, the point is that data falling in this SZA value range are not used in level 3.

The categorization of CLAAS-3 data into day and night retrievals (mentioned in line 107) specifically refers to level 3 data, since level 2 data are produced on a 15-minute basis. This point will also be clarified in the revised manuscript.

*Line 143: so does this new cloud mask algorithm actually perform better for marine stratus off Africa at sunrise and sunset? Or the issue is still there and that is why you're cutting out your solar zenith that way?*

We don't understand this comment. Line 143 and the relevant paragraph provide a short description of the probabilistic cloud mask algorithm used in CLAAS-3, without referring to specific regions or illumination conditions. Relevant evaluation results are shown in Section 4.1, and a detailed assessment of the product is given in the CLAAS-3 validation report (CM SAF, 2022d).

*Line 151: Your training dataset is based on CALIOP, which means \*all\* your training data is early afternoon. Yet you apply the algorithm to all observation times. How does that affect the accuracy of your neural network? What are your uncertainties outside of 1:30pm local time? You say that you calculate them, but what kind of value ranges are you getting?*

Yes, this is a limitation of the CALIOP observations. However, don't forget that also nighttime conditions (at 1:30 am) are covered. Nevertheless, we claim that the consequences are not serious. This is mainly deduced from our results from validation studies using other kind of data (e.g., including both MODIS and surface observation datasets) as reported in the extensive validation report. Actually, a good example is provided in Figure 7-6 in the validation report, also shown below.

[Figure]

Validation Report Figure 7-6: Maps of averaged cloud fractional cover based on the overlapping time period of CLAAS-3 and CALIOP L3 (2006/06 – 2016/12) from CLAAS-3 (left column), top layer flavor of CALIPSO-GEWEX cloud product (middle column) and their difference (right column).

Here we have a comparison between monthly means based on CALIOP observations (i.e., polar data in fixed sun synchronous orbit providing only 2 observations per day with a coarse spatial sampling) and monthly means based on CLAAS-3 observations (utilizing observations every 15 minutes). Despite

the huge difference in temporal and spatial sampling the results are surprisingly similar. Only in the tropical region we see a clear difference and this is simply explained by the higher sensitivity of very thin cirrus clouds for CALIOP which are abundant in the tropical region and largely subvisible for the SEVIRI sensor. The same pattern of differences appears in Figure 2 of the manuscript, where the level 2 matchups from CLAAS-3 and CALIOP are representative of the CALIOP overpass time only. Thus, we see no strong signal that the differences in the temporal sampling lead to big problems. Notice that we also compared with many other datasets (including MODIS, another polar dataset with fixed observation times) which basically gives the same results (although not providing exactly identical results, mostly depending on differences in algorithms and the used spectral bands).

So, how is this possible? The reason is that the cloud detection method is organized in a way that it doesn't depend strictly on the exact local observation time. Rather, we use the data to characterize mean daytime and mean nighttime conditions separately but that we also separate these results over several Earth surface categories. Notice also that visible measurements are translated into normalized reflectances (i.e., referring to conditions with an overhead sun) which reduces the dependence on the exact solar zenith angle or local time. However, we must correct the Referee in that the method is not based on the training of a neural network. Instead, we are using a Naïve Bayesian method trained with CALIOP cloud masks. We characterize the dependency of the CALIOP cloud mask on a certain set of SEVIRI image spectral bands or derived image features (details explained in Karlsson et al., 2020). The training also takes into account surface temperatures and total atmospheric moisture contents (taken from ERA-5 reanalysis data) and surface emissivities (MODIS-based). The use of total atmospheric moisture contents is also tightly linked to existing viewing angles (i.e., corrections increase with increasing viewing angles) which implicitly accounts to some extent for another limitation of CALIOP observations, namely the fixed near-nadir observation mode.

In conclusion, we don't see any obvious problems with our approach that could make results less useful for other observation times and viewing angles than the fixed CALIOP local times and viewing angles. Observe that we do not deny that there might be problems related to the sampling differences but we cannot currently see that these have a strong impact on results. To really pinpoint how any potential limitation really looks outside the 1:30 pm observation time is actually not easy since there is not any reference that can give CALIOP-quality cloud information with high resolution globally and at any time.

In the revised manuscript we will clarify our way to generalize the problem to mean daytime and nighttime conditions and the use of additional information to implicitly account for varying viewing angles. Regarding the calculation of uncertainties, cloud mask uncertainties (level 2) are defined from the calculated cloud probabilities (based on training data). Maximum uncertainty is reached for cloud probability 50 % while highest confidence is given at values 0 % and 100 %. The user can get access to this information in level 2 files. For the level 3 product, a simple estimation based on the averaging of the probability distance from the 50 % threshold for clear and cloudy pixels is provided.

*Line 175: Your cloud microphysics retrievals methods are eerily similar to SEV06-CLD. I am wondering why that product is not mentioned in any way.*

The list of similar datasets mentioned in the paper is not exhaustive. We have focused on referring only to operational, documented, and validated datasets, especially for those used to evaluate CLAAS-3. Additionally, datasets used as reference for the evaluation should be as independent from CLAAS-3 as possible. Since SEV06-CLD is based on SEVIRI, just as CLAAS-3,

it is not a very independent dataset. Nevertheless, we acknowledge the Referee's point regarding its relevance, and we will include a mention of it in Section 2.1.

*Line 194: replace "… on board the Cloud-Aerosol Lidar and Infrared Pathfinder Satellite Observation (CALIPSO) satellite" with "on board the Cloud-Aerosol Lidar and Infrared Pathfinder Satellite Observation (CALIPSO) spacecraft". Repetitive.*

Will be done.

*Line 220: could you just drop some ballpark uncertainty values that you encounter with CALIOP? For more discussion, of course the reference, but a few numbers here would do some good.*

The CALIOP cloud datasets have undergone several revisions and we assume that the Referee asks for the accuracy of the latest revision (CAL_LID_L2_05kmCLay200 Standard-V4-20) which we have used for training and validation of the cloud mask, cloud top height and cloud phase products. The official information about the quality of this version and improvements compared to earlier versions is provided in: https://www-calipso.larc.nasa.gov/resources/calipso_users_guide/qs/cal_lid_l2_all_v4-20.php. This webpage contains detailed information on various technical aspects, e.g. how data calibration has improved, use of an improved surface detection, improved cloud-aerosol layer separation and discrimination and cloud subtyping improvements. To understand how all this affects the final cloud products is not easy and is not quantified in this website. We are not aware of any other publication that has provided this kind of information. However, the most prominent change for us has been adjustments of the cloud optical thickness (COT). Previous estimations were too low and the new COT values have increased by up to 25 % for the thinnest clouds. This has clearly changed (hopefully to the better) all operations where we have filtered our results (i.e., removing impact of the thinnest CALIOP-detected clouds, like in Figures 3 and 8).

Regardless of this lack of detailed quality information of the end cloud products from CALIOP, we are confident about their high quality. Simply the fact that this sensor measures backscatter from cloud particles and not (as for passive imagers) radiation from all sorts of emitters/reflectors in addition to the radiation from clouds, makes the access to these observations extremely valuable for everybody working with cloud retrievals from passive imagery. More important is actually the collocation problem, i.e., the actual comparison in space and time between CALIOP cloud products and SEVIRI (and any other imaging sensor) data which is fundamental for the training process. Notice that the actual core FOV for CALIOP is about 70 meters in size in comparison to typical image data with scales of several km. It means that for clouds with scales less than a few km we can find considerable deviations between the two datasets (i.e., CALIOP could easily report a cloud while the observation from an imager indicates cloud-free, and vice-versa). For larger cloud scales problems are small (except for that various cloud properties can still differ also for larger scale clouds). Also this 'noise' in the collocation process is difficult to quantify exactly but it clearly leads to the fact that we can never reach 100 % agreement in the measured cloud amounts from the two sensors. Measures like the probability of detection (POD) normally saturates at values close to 95 %, thus never reaching 100 %. There will always be cases when small scale clouds (or small scale holes in clouds) are missed by the imaging instrument in the collocation process. To this comes also effects of time differences in the observations. A larger time difference than 3 minutes normally increases these deviations.

In conclusion, we think that CALIOP cloud information is more than sufficient when it comes to accuracy for our applications. A bigger problem is the collocation errors which put some limit on what really can be achieved, especially in areas with a high frequency of broken or fractional clouds.

We will clarify some of these aspects in the revised manuscript.

*Line 260: If you would've looked at SEV06-CLD product that is the MODIS C6.1 code running on SEVIRI, you would have an exact uncertainty comparison due to instrument differences rather than waving of hands you are doing here.*

We assume the Referee refers to the paragraph in lines 263-266, where MODIS retrieval uncertainties are mentioned. The MODIS C6.1 algorithms use additional channels (e.g., four $CO_2$ channels to determine cloud top pressure, height and temperature: Baum et al., 2012) compared to what is available on SEVIRI. Therefore, also the SEV06-CLD cannot use these. These additional channels are expected to lower the retrieval uncertainty to some extent, which motivates the qualitative statement given in this paragraph.

Baum, B. A., W. P. Menzel, R. A. Frey, D. C. Tobin, R. E. Holz, S. A. Ackerman, A. K. Heidinger, and P. Yang, 2012: MODIS Cloud-Top Property Refinements for Collection 6. J. Appl. Meteor. Climatol., 51, 1145–1163, https://doi.org/10.1175/JAMC-D-11-0203.1.

*Line 320: CER actually is a rather mixed bag, heavily influenced by above-cloud aerosol. You can't really make a statement that cloud drops get larger towards the cloud top. Bennartz and Rausch are careful to indicate that it's not always true. Moreover if there is any ice cloud in the scene at all, 3.7um will always be smaller than 1.6um. Anyways, it's more complicated. I don't know what impact if any that has on your conclusions, but still.*

We do not intend to claim that this statement is always true. But indeed this phrasing can be misunderstood, and it will be revised in the updated manuscript. The second bullet point will be changed to: "Only cases where CRE retrieved at 3.9 µm is greater than CRE retrieved at 1.6 µm were considered. This criterion is also applied in the Bennartz and Rausch (2017) CDNC data set based on MODIS. It is meant to select idealized stratiform boundary layer clouds (ISBLC), as they are termed in that study. However, several other factors, including cloud inhomogeneity, and the presence of thin cirrus, can occur, impacting the retrievals such that this inference of ISBLC may not be correct."
As explained in the same study, however, this criterion is not useful for rejecting above-cloud aerosol cases. We use the AAI threshold to exclude such cases.

*Line 335: Again, that is a very old study that was moreover influenced by outright bugs in the 3.7um retrieval algorithm at the time. The issues were not corrected until Data Collection 5 was released. Please don't quote that paper. What impact this has on your use of DARDAR data, I couldn't tell you, but I would not base a single thing on that paper, particularly because you're making statements about penetration depth and retrieval values. Retrieval values were garbage, differences in penetration depth completely aside.*

Does the reviewer refer to Platnick (2000), mentioned in line 336? Unfortunately, we omitted this paper from the References section. The reference is given below. This study is mentioned here as a reference on the usage of vertical weighting functions for the estimation of 'neartop' CRE, to reflect the relevant vertical sensitivity of the retrieval. In the case of DARDAR data, using the vertical weighting function which emphasizes the cloud top led to a decrease in bias, compared to the usage of a vertically uniform weighting function. However, the bias remains considerable (Section 4.5).

Platnick, S.: Vertical photon transport in cloud remote sensing problems, J. Geophys. Res. Atmos., 105, 22919–22935, https://doi.org/10.1029/2000jd900333, 2000.

*Line 385: Can you show an image same as figure 2, that the bias actually decreases with additional screening criteria applied? That -30%. Oooof.*

In Figure 2 of the revised manuscript we will include maps of CALIOP CFC and corresponding CLAAS-3 – CALIOP differences using COT > 0.1 as CALIOP cloud detection criterion, in addition to the COT > 0 already used. There is an overall decrease in negative biases with the latter criterion. As a lateral consequence, there is also a tendency of increase in positive biases. The new figure is shown below.

[Figure]

**Figure 1: Spatial distribution of the 2013 average level 2 Cloud Fractional Coverage (CFC) from CLAAS-3 (a), CALIOP (b), and their difference (c). Here CFC is estimated from the binary cloud mask by collecting matchups to a regular 1.5° × 1.5° grid and averaging them within each grid box. CALIOP cloud detection criterion is total column COT > 0 (b and c) and COT > 0.1 (d and e). A 2-dimensional Gaussian filtering was used for noise reduction.**

*Figure 4: It seems from this image that CLAAS-3 is picking up persistent aerosol loads as clouds. It is very dusty in the areas with the highest positive biases. Can you explain?*

We don't have the same impression, but this possibility cannot be excluded. Specifically, high positive biases indeed prevail over the Arabian Peninsula and Iran, where dust aerosol loads

are high. However, in northwestern Africa, where dust loads are also high, mainly negative biases are found, suggesting possible combined effects that include different viewing angles. We have not investigated this further. Please note also that no CLAAS-3 overestimation compared to CALIOP is present in the Middle East (Figure 2).

*Figure 11: Don't tell me you used Data Collection 5 MODIS data! For Collection 6.1 each retrieval pair has its own optical thickness reported. This kind of difference in count between tau and re was only present in C5 data and older. Please explain.*

As stated in the relevant section (3.3), all MODIS data used in this study are from Collection 6.1. The difference in the number of collocations appearing between COT and CRE originates in CLAAS-3, not in MODIS. In CLAAS-3 retrievals, cases of reflectance pairs lying below or above the Nakajima-King LUT lead to successful retrievals of COT but failed retrievals of CRE (and consequently CDNC and CGT). We will revise Figure 11 caption to clarify this point.

*Figure 14: Same thing. What data collection did you use here? This doesn't look right at all, if you used C6.1. Number of successful retrievals for optical thickness and effective radius is identical and each band combination has its own tau, re and water path. They've been entirely decoupled in the latest MODIS data collection.*

Please see our reply above. The Figure 14 caption will also be updated accordingly in the revised manuscript.